# SARS-CoV-2 requires cholesterol for viral entry and pathological syncytia formation

**David W Sanders[1†], Chanelle C Jumper[1†], Paul J Ackerman[1†], Dan Bracha[1], Anita Donlic[1], Hahn Kim[2,3], Devin Kenney[4,5], Ivan Castello-Serrano[6], Saori Suzuki[7], Tomokazu Tamura[7], Alexander H Tavares[5,8], Mohsan Saeed[5,8], Alex S Holehouse[9], Alexander Ploss[7], Ilya Levental[6], Florian Douam[4,5], Robert F Padera[10], Bruce D Levy[11], Clifford P Brangwynne[1,12]***

[1]Department of Chemical and Biological Engineering, Princeton University, Princeton, United States; [2]Princeton University Small Molecule Screening Center, Princeton University, Princeton, United States; [3]Department of Chemistry, Princeton University, Princeton, United States; [4]Department of Microbiology, Boston University School of Medicine, Boston, United States; [5]National Emerging Infectious Diseases Laboratories, Boston University, Boston, United States; [6]Department of Molecular Physiology and Biological Physics, University of Virginia, Charlottesville, United States; [7]Department of Molecular Biology, Princeton University, Princeton, United States; [8]Department of Biochemistry, Boston University School of Medicine, Boston, United States; [9]Department of Biochemistry and Molecular Biophysics, Washington University School of Medicine, St. Louis, United States; [10]Department of Pathology, Brigham and Women's Hospital and Harvard Medical School, Boston, United States; [11]Pulmonary and Critical Care Medicine, Brigham and Women's Hospital and Harvard Medical School, Boston, United States; [12]Howard Hughes Medical Institute, Princeton, United States

***For correspondence:**
cbrangwy@princeton.edu

[†]These authors contributed equally to this work

**Abstract** Many enveloped viruses induce multinucleated cells (syncytia), reflective of membrane fusion events caused by the same machinery that underlies viral entry. These syncytia are thought to facilitate replication and evasion of the host immune response. Here, we report that co-culture of human cells expressing the receptor ACE2 with cells expressing SARS-CoV-2 spike, results in synapse-like intercellular contacts that initiate cell-cell fusion, producing syncytia resembling those we identify in lungs of COVID-19 patients. To assess the mechanism of spike/ACE2-driven membrane fusion, we developed a microscopy-based, cell-cell fusion assay to screen ~6000 drugs and >30 spike variants. Together with quantitative cell biology approaches, the screen reveals an essential role for biophysical aspects of the membrane, particularly cholesterol-rich regions, in spike-mediated fusion, which extends to replication-competent SARS-CoV-2 isolates. Our findings potentially provide a molecular basis for positive outcomes reported in COVID-19 patients taking statins and suggest new strategies for therapeutics targeting the membrane of SARS-CoV-2 and other fusogenic viruses.

## Introduction

COVID-19 has caused over a million deaths in the year following identification of its causative pathogen, severe acute respiratory syndrome coronavirus 2 (SARS-CoV-2) (*Zhu et al., 2020a*). Building on knowledge of similar enveloped coronaviruses (*Belouzard et al., 2012*; *Heald-Sargent and Gallagher, 2012*), recent studies made astonishing progress toward a holistic understanding of SARS-CoV-2 pathobiology, suggesting amenable targets to therapeutic intervention (*Haynes et al., 2020*;

*Stratton et al., 2021*; *Tay et al., 2020*). In particular, the unprecedented pace of SARS-CoV-2 research led to key insights into viral fusion (*V'kovski et al., 2021*). Many early studies, including most small molecule and genetic screens, focused on entry (*Chen et al., 2020*; *Dittmar et al., 2020*; *Riva et al., 2020*; *Wei et al., 2020*; *Zhu et al., 2020b*). Central to these efforts are trimeric spike glycoproteins (or 'peplomers'), which give the viral envelope its crown-like appearance, and ACE2, their essential human receptor (*Duan et al., 2020*; *Mittal et al., 2020*). Association of the two proteins underlies virus-cell adhesion, which precedes a conformational change in spike that unleashes its fusion machinery to infiltrate the cell (*Hoffmann et al., 2020a*; *Hoffmann et al., 2020b*; *Ke et al., 2020*; *Lan et al., 2020*; *Shang et al., 2020*; *Wrapp et al., 2020*; *Yan et al., 2020*).

While essential protein-protein interactions for infectivity have been forthcoming, equally important aspects of SARS-CoV-2 pathobiology have received less attention. For example, amplification of systemic infection requires mass production of functional virions, each of which relies on a specific set of biomolecules to orchestrate the optimal number and spacing of spike trimers in its envelope (*Ke et al., 2020*; *Klein et al., 2020*). How assembly occurs efficiently in the crowded cellular environment is unclear. One favored hypothesis is that viral proteins are similarly trafficked to the ER-Golgi intermediate compartment (ERGIC). While cargo receptor-binding likely plays a role, an alternative possibility is that such proteins feature an intrinsic preference for membrane domains of distinct lipid composition (*Cattin-Ortolá et al., 2020*; *Li et al., 2007*; *Liao et al., 2006*; *Lu et al., 2008a*; *McBride et al., 2007*; *Thorp and Gallagher, 2004*). Indeed, certain viruses require association between receptor proteins and specific lipids to trigger endocytosis (*Levental et al., 2020*; *Pelkmans, 2005*). Whether this is the case for ACE2 remains to be determined. Regardless, lipid bilayers' differential propensity to incorporate spike vs. ACE2 might determine whether premature interactions promote unproductive membrane fusion in the cell interior, or if present at the cell surface, fusion of apposing cells (*Buchrieser et al., 2020*; *Cattin-Ortolá et al., 2020*; *Li et al., 2003*; *McBride and Machamer, 2010a*; *Ou et al., 2020*; *Papa et al., 2020*; *Xia et al., 2020*).

For many enveloped viruses, infection indeed causes fusogenic viral protein display on the host cell plasma membrane, which allows neighboring cells to fuse into multinucleated 'syncytia' (*Ciechonska and Duncan, 2014*; *Compton and Schwartz, 2017*; *Duelli and Lazebnik, 2007*). Past studies of respiratory syncytial virus (RSV), human immunodeficiency virus (HIV), and others suggest that cell-cell fusion can play key roles in pathogenicity, whether it be in viral replication, or evasion of the host immune response (*Frankel et al., 1996*; *Johnson et al., 2007*; *Maudgal and Missotten, 1978*). Pioneering work on SARS-CoV-1 (*Li et al., 2003*) as well as recent studies on SARS-CoV-2 identified similar syncytia (*Buchrieser et al., 2020*; *Cattin-Ortolá et al., 2020*; *Hoffmann et al., 2020a*; *Ou et al., 2020*; *Papa et al., 2020*; *Xia et al., 2020*; *Zang et al., 2020b*), which may or may not be relevant to patient pathology (*Bryce et al., 2020*; *Giacca et al., 2020*; *Rockx et al., 2020*; *Tian et al., 2020*). It remains an open question if syncytia are related to viral and host cell membrane composition, and whether their formation provides mechanistic insights into cholesterol-targeting therapeutics repurposed for COVID-19 treatment (*Daniels et al., 2020*; *Zhang et al., 2020*).

Here, we address these significant gaps in our understanding of COVID-19 pathobiology by employing a suite of microscopy-based approaches built around the finding that co-cultures of ACE2- and spike-expressing cells amass widespread syncytia. Mechanistically, ACE2-spike clusters assemble at transcellular, synapse-like contacts, which precede fusion pore formation and multinucleation. A high-throughput screen for modulators of cell-cell fusion, involving ~6000 compounds and >30 spike variants, collectively underscore an essential role of biophysical features of the membrane, particularly spike-associated cholesterol, for SARS-CoV-2 infection. Our results suggest that modulation of membrane composition may inhibit viral propagation, and further informs critical lipid-protein assemblies in physiological syncytia and cell adhesion.

## Results

### Syncytia derive from fusion events at synapse-like, spike-ACE2 protein clusters

Given the central role of the ACE2-spike interaction in viral infection (*Hoffmann et al., 2020b*; *Li et al., 2003*; *Mittal et al., 2020*), we sought to develop a live cell microscopy assay of binding and membrane fusion. We generated pooled populations of human osteosarcoma (U2OS) cells,

chosen for their flat morphology and lack of critical fusion machinery (*Beck et al., 2011*), which stably express fluorescently tagged ACE2 or spike (full-length, 'FL' vs. receptor-binding domain, 'RBD'; see *Figure 1A* for domain organization), using the B7 transmembrane ('TM') domain (*Liao et al., 2001*; *Lin et al., 2013*) as a control. Upon co-culture, ACE2 and spike RBD cluster at cell-cell interfaces in a binding-dependent manner (*Figure 1B*). By contrast, and in agreement with others (*Buchrieser et al., 2020*; *Cattin-Ortolá et al., 2020*; *Hoffmann et al., 2020a*; *Ou et al., 2020*; *Xia et al., 2020*; *Zang et al., 2020b*), spike FL/ACE2 interactions drove membrane fusion, with the vast majority of cells joining multinucleated syncytia after a day of co-culture (*Figure 1C*).

We reasoned that if this co-culture system recapitulates established findings regarding spike/ACE2-mediated viral entry, it might serve as a useful high-throughput proxy assay for infection, without need for enhanced biosafety protocols. To examine this possibility, we first confirmed that fusion events occur following co-culture of spike cells with infection-competent cell lines (VeroE6, Calu3) in absence of ACE2 overexpression, but not with those that do not support infection (Beas2B, U2OS without ACE2) (*Figure 1—figure supplement 1A*; *Hoffmann et al., 2020b*). We further validated the relevance of the assay by showing that domains required for virus-cell entry (e.g. binding domain: RBD; fusion machinery: heptad repeats and fusion peptide) are needed for cell-cell fusion (*Figure 1—figure supplement 1B*). Similar to results obtained with infectious virus (*Hoffmann et al., 2020a*), fusion required the spike S2' cleavage site but not the S1/S2 site (*Figure 1—figure supplement 1C*). Finally, different fluorescent tags (GFP, mCherry, iRFP) gave similar results (*Figure 1—figure supplement 1D*) expanding the fluorescent toolkit for live cell studies.

We hypothesized that spike/ACE2-mediated syncytia form in a stepwise manner, which might illuminate mechanisms of formation and pathogenesis. We performed live cell microscopy of co-cultures, documenting dozens of fusion events preceding large syncytia. Invariably, contact between opposite cell types (spike vs. ACE2) results in near instantaneous accumulation of spike protein clusters at ACE2-containing membrane protrusions (*Figure 1D–E*). These punctate structures are long-lived (minutes) (*Figure 1-video 1*), similar to physiological synapses (e.g. neuronal, immunological) (*Cohen and Ziv, 2019*; *Dustin, 2014*). In all observed cases, fusion events proceed from such synapses (*Figure 1E–F*; *Figure 1—video 2*), often within a few minutes of their formation, but frequently following longer durations of time. In most (but not all) examples, fusion pore dilation follows retraction of an individual spike cluster toward the interior of an ACE2 cell (*Figure 1F–I*), suggesting that motility-associated mechanical forces (e.g. actomyosin contractility) and/or endocytosis is pivotal to overcoming the energetic barrier to lipid bilayer mixing. When cells are plated at high density, most 'primary' fusion events occur within 60 min (*Figure 1G–J*), with latter 'secondary' amalgamation of small syncytia into progressively larger structures (*Figure 1K*; *Figure 1—video 3*). Over time, syncytia undergo vacuolization, likely from fusion-driven collapse of intracellular organelles into hybrid compartments (*Figure 1L*). By 48–72 hr, cells disintegrate into immobile spike/ACE2-coated vesicles, having eaten themselves from within (*Figure 1M*).

## Syncytia are a defining pathological feature of COVID-19

While clearly useful for interrogating spike domains that modulate membrane fusion, a critical gap in our knowledge concerns the pathophysiological relevance of the syncytia themselves. Given the cytotoxic consequences of cell fusion (*Figure 1M*), an appealing hypothesis is that ACE2/spike-mediated cell vacuolization contributes in part to the diffuse alveolar damage observed in the lungs of COVID-19 patients (*Menter et al., 2020*). Intriguingly, SARS-CoV-2 spike is a particularly potent mediator of syncytia formation relative to both SARS-CoV-1 spike and commonly studied fusogens (e.g. p14 FAST, MYMK/MYMX) (*Bi et al., 2017*; *Chan et al., 2020b*) based on side-by-side comparisons of cell populations with similar expression levels (*Figure 2A–C*; see Materials and methods for details on expression measurements).

We speculated that this superior ability to promote cell-cell fusion might be reflected by unique pathological attributes in vivo. If so, we predicted that syncytia would be readily detected in the lungs of COVID-19 patients. To test this, we histologically evaluated lung samples from 24 deceased patients ('decedents') with diffuse alveolar damage secondary to SARS-CoV-2 infection; six decedents who were positive for SARS-CoV-2 but did not have pulmonary manifestations (died of other causes); and nine control decedents with diffuse alveolar damage (died prior to SARS-CoV-2 discovery in 2019). Multinucleated syncytia were detected in 18 of 24 decedents who died as a direct consequence of SARS-CoV-2 infection (*Figure 2D–H*), a finding supported by other patient cohorts

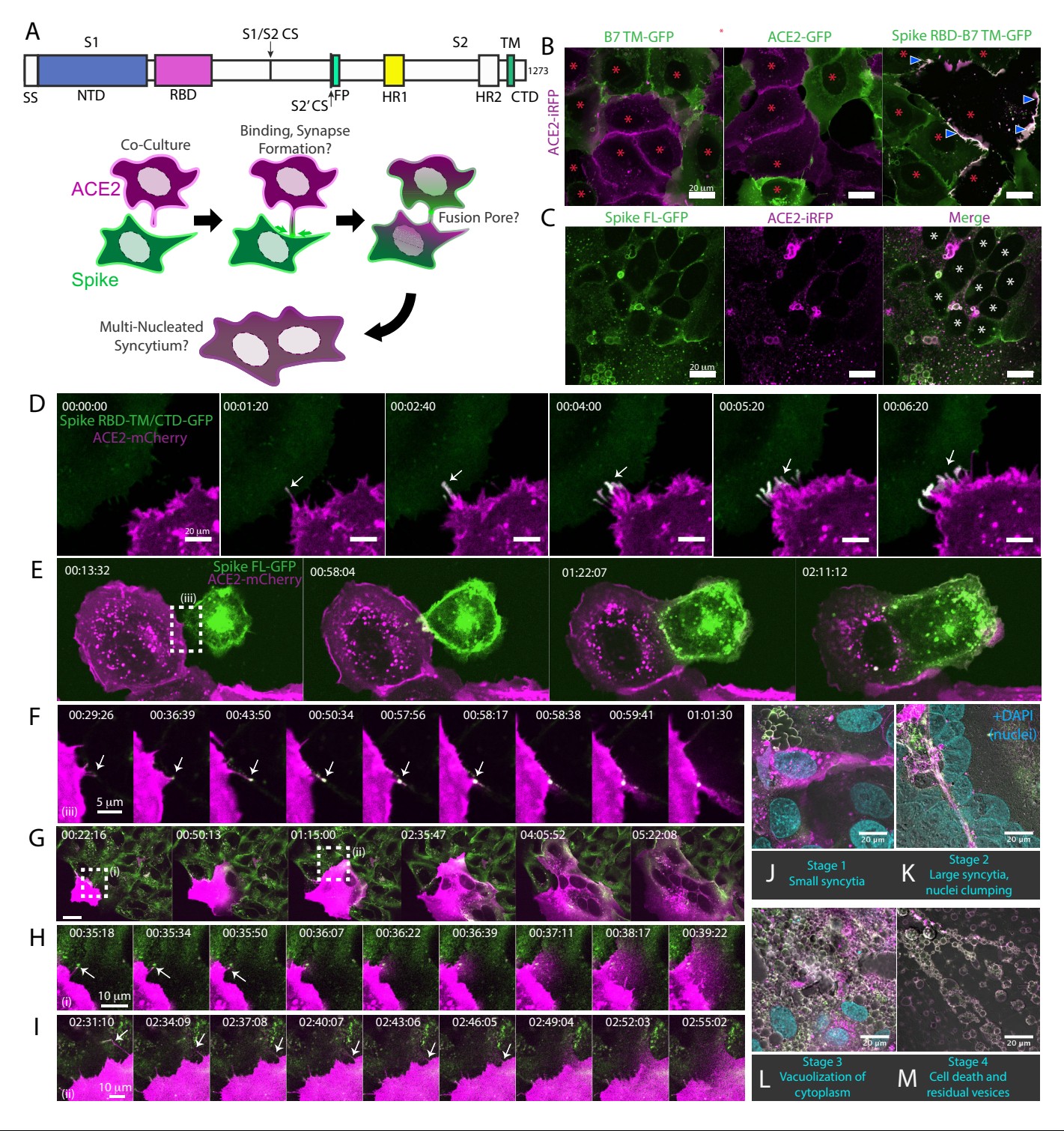

**Figure 1.** Syncytia derive from fusion events at synapse-like, spike-ACE2 protein clusters. (**A**) Top: Domain structure of a single monomer of the SARS-CoV-2 spike trimer. Domains/motifs is from left to right (see KEY RESOURCES table for residue boundaries): SS (signal sequence), NTD (N-terminal domain), RBD (receptor-binding domain), S1/S2 (subdomain-1/2 cleavage site), S2' (subdomain-2' cleavage site), FP (fusion peptide), HR1 (heptad repeat 1), HR2 (heptad repeat 2), TM (transmembrane alpha helix), CTD (cytoplasmic domain). Bottom: Cartoon depiction of live cell co-culture assays to detect spike-ACE2 binding and cell-cell fusion. Magenta, acceptor cells (human ACE2-mCherry or ACE2-iRFP); Green, donor cells (GFP-tagged spike variant). (**B**) ACE2-iRFP U2OS (human osteosarcoma) acceptor cells (magenta) co-cultured for 24 hr with U2OS cells expressing GFP-tagged proteins (green): B7 transmembrane (TM, left), ACE2 (middle), spike receptor-binding domain (RBD-TM/CTD, right). Red asterisks indicate single cell nuclei in

*Figure 1 continued on next page*

*Figure 1 continued*

isolation (no syncytia); arrowhead, synapses (select examples noted). (C) Co-culture of U2OS acceptor cells expressing ACE2-iRFP (magenta) with spike full-length ('FL')-GFP U2OS cells (green). White asterisks: cell nuclei in syncytium. (D) Co-culture of ACE2-mCherry (magenta) and spike RBD-TM/CTD-GFP (green) cells for indicated amount of time (hours:minutes:seconds). Arrow: synapse-like interfaces between cells. Scale-bar, 5 µm. See also *Figure 1—video 1* for long-lived ACE2-Spike FL synapses. (E) Similar to (D), but spike FL-GFP and ACE2-mCherry co-culture. Dashed box indicates site of synapse formation and cell-cell fusion. See *Figure 1—video 2* for time-lapse movie. (F) Zoom-in on synapse formation (arrow, left image) and fusion event (arrow, sixth image from left) of dashed box in (E). (G) ACE2-mCherry cell added to pre-plated spike FL-GFP U2OS cell monolayer (time since ACE2 cell plating indicated). Syncytium forms by multiple cell-cell fusion events (dashed boxes). See (H,I) for zoom-in events (i) and (ii). See *Figure 1—video 3* for time-lapse movie. (H) First cell fusion event (i from G) at spike-ACE2 synapse. Time since ACE2-mCherry cell plating indicated. Arrow: retracting synapse prior to cell fusion. (I) Similar to (H) but second cell-cell fusion event (ii from G). (J) Representative image of small syncytia (stage 1) common at early time points following co-culture of ACE2-mCherry (magenta) and spike-GFP (green) U2OS cells but rare at 24 hr (blue, Hoechst DNA stain). (K) Similar to (J), but representative of more common, larger syncytia (stage 2) at 24 hr. Nuclei (blue) clump in center of syncytium. (L) Similar to (J), but representative of typical syncytium with extensive vacuolization (stage 3) at 48 hr. (M) Similar to (J), but representative of remnants (spherical membranous structures) of dead syncytium at 72 hr (stage 4). See also *Figure 1—figure supplement 1*; *Figure 1—video 1–3*.

**Figure 1—video 1.** Transcellular ACE2-spike synapses are long-lived cellular assemblies.

https://elifesciences.org/articles/65962#fig1video1

**Figure 1—video 2.** ACE2-spike synapse formation and cell-cell fusion following co-culture.

https://elifesciences.org/articles/65962#fig1video2

**Figure 1—video 3.** Building a syncytium: multiple cell-cell fusion events following addition of a single ACE2 cell to a spike cell monolayer.

https://elifesciences.org/articles/65962#fig1video3

**Figure supplement 1.** Syncytia derive from fusion events at synapse-like, spike-ACE2 protein clusters. (A) Indicated non-transduced cells (or ACE2-mCherry/U2OS control) co-cultured with U2OS spike-GFP (green) cells for 24 hr. White asterisks indicate nuclei in syncytia; red, in isolation. (B) Indicated GFP-spike variant (green) U2OS cells co-cultured with U2OS ACE2-mCherry (magenta) cells for 24 hr. White asterisks indicate nuclei in syncytia; red, in isolation; arrowhead, synapses (select examples noted). (C) Similar to (B), but using spike variants that disrupt its two cleavage sites (S1/S2 vs. S2'). (D) U2OS cells expressing spike or ACE2 with indicated fluorescent tag, co-cultured for 24 hr. White asterisks indicate nuclei in syncytia; red, in isolation; arrowhead, synapses (select examples noted).

(*Giacca et al., 2020*; *Tian et al., 2020*). These syncytia were of lung epithelial origin, as demonstrated by nuclear staining for TTF-1 (NKX2-1) (*Figure 2F*). In contrast, only one of the nine decedents with diffuse alveolar damage from other causes demonstrated multinucleated syncytia, indicating that these syncytia are not a common feature of lung inflammation (*Figure 2G,H*). They were also absent in lung tissue from the six SARS-CoV-2 decedents who did not show pulmonary manifestations and died of other causes. Thus, pathological syncytia are a direct consequence of pulmonary involvement by SARS-CoV-2 (*Figure 2H*). These syncytia, however, were generally not positive for the SARS-CoV-2 nucleocapsid protein, similar to previous reports (*Bryce et al., 2020*; *Rockx et al., 2020*). Thus, we cannot rule out a yet-to-be identified pulmonary abnormality specific to SARS-CoV-2 infection (but spike-independent), or related to free spike proteins in the alveolar debris, in pathological syncytia formation.

## A novel high-throughput screening platform identifies modulators of syncytia formation

Given the potential pathological relevance of syncytia and their ability to interrogate SARS-CoV-2 entry, we sought to adapt our cell model into a high-throughput assay to uncover molecular mechanisms and drug targets. We developed and evaluated three different fixed-cell microscopy assays, each of which used fluorescent proteins as readouts for fusion (*Figure 3—figure supplement 1B,D*), with total nuclei number serving as a toxicity measure. Two of these assays (human U2OS-ACE2 vs. monkey VeroE6 heterokaryon) leveraged RNA-binding proteins' ability to shuttle between the nucleus and cytosol (*Iijima et al., 2006*; *Zinszner et al., 1997*), with nuclear co-localization of mCherry/GFP reflecting cell fusion. The third assay used split-GFP (*Buchrieser et al., 2020*; *Feng et al., 2017*), which only fluoresces when its two halves come into contact (e.g. after fusion). After careful assessment, the U2OS-ACE2 heterokaryon system was shown to be the superior assay based on its Z'-factor (0.85), a measure of separation between positive (spike RBD/ACE2 co-culture, no fusion) and negative (spike FL/ACE2 co-culture) controls (*Figure 3A*, *Figure 3—figure supplement 1D*); generally Z'-factor > 0.5 is considered excellent for a high-throughput assay (*Zhang et al., 1999*). To determine an optimal time point for quantification, co-cultures were

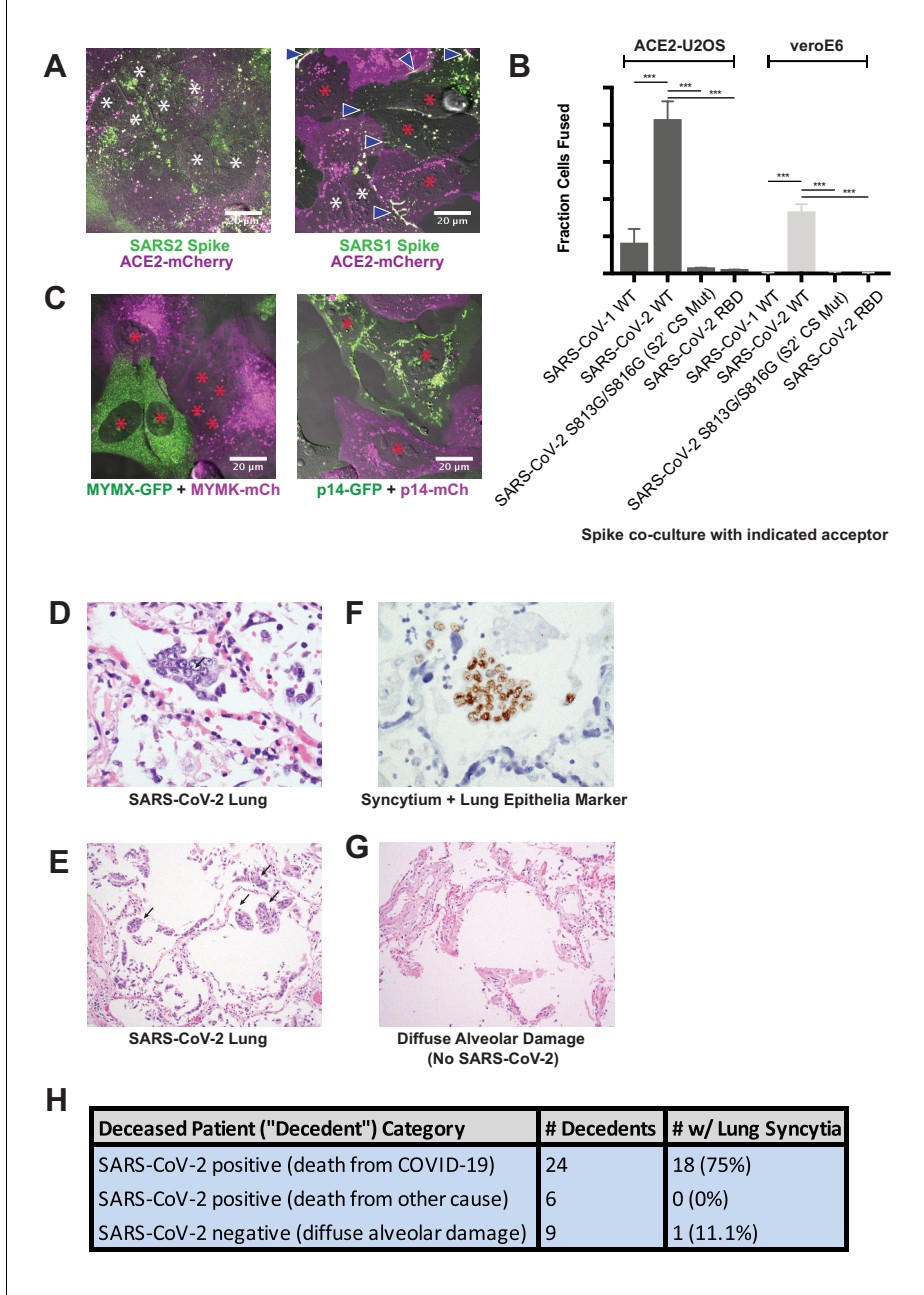

**Figure 2.** Syncytia are a defining pathological feature of COVID-19. (A) ACE2-mCherry (magenta) U2OS cells co-cultured with GFP-tagged (green) SARS-CoV-2 spike cells (left) or SARS-CoV-1 spike cells (right) for 24 hr. White asterisks indicate nuclei in syncytia; red, in isolation; arrowhead, synapses (select examples noted). (B) Quantification of (A) by percent cells fused (also tested VeroE6 donor cells, which express endogenous ACE2, and control SARS-CoV-2 spike variants that lack ability to promote fusion). Mean and SEM: n = 4 biological replicates (16 images per). p-Values of <0.01, <0.001, and<0.0001 are represented by *, **, and ***, respectively. (C) Indicated fusogen-expressing cells lines co-cultured for 24 hr. Red asterisk indicates nuclei of single cells (not in syncytia). (D) Lung from SARS-CoV-2 decedent demonstrating syncytia formation (H and E stained section, ×400 original magnification). Syncytium labeled with arrow. (E) Similar to (D), but sample obtained from different deceased COVID-19 patient and at ×100 magnification. Syncytium labeled with arrow. (F) Immunohistochemistry for lung epithelia marker TTF-1 (NKX2-1; brown) showing nuclear positivity in the syncytial cells (×400 original magnification). (G) Lung from control decedent with diffuse alveolar damage unrelated to SARS-CoV-2 infection (died pre-2019), showing hyaline membranes (remnants of dead cells; bright pink) but no syncytia (H and E stained section, ×100 original magnification). (H) Table summarizing decedents examined for syncytia pathology.

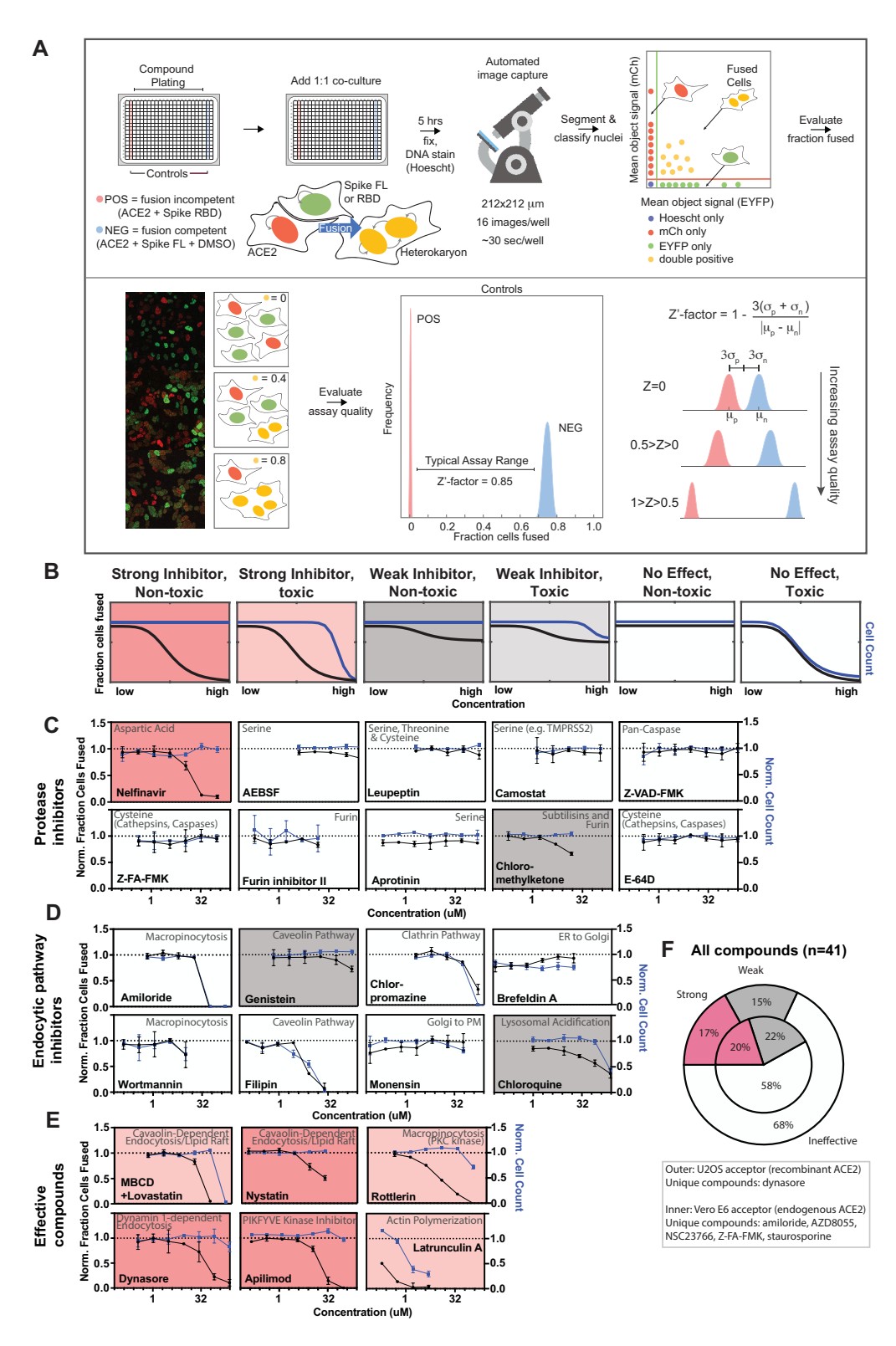

**Figure 3.** A novel high-throughput screening platform identifies modulators of syncytia formation. (A) Heterokaryon assay workflow overview (top) and characterization (bottom). Equal parts acceptor cells (express ACE2-iRFP and FUS-mCherry) plus donor cells (express spike FL-iRFP and HNRNPA1-EYFP) are co-cultured in 384-well microtiter plate: positive control (spike RBD, red column), negative control (DMSO, blue column), test compounds (other columns). After 5 hr, cells are fixed and nuclei are stained (Hoechst) then identified/segmented by automated confocal microscopy. Fraction cells

*Figure 3 continued on next page*

*Figure 3 continued*

fused is determined by percent co-positive (mCherry and EYFP) nuclei. Sample images and schematic interpretation of a positive control well (top), test well with reduced fusion (middle), negative control well (bottom). Z'-factor measures window size with higher score indicating a more robust screening platform. (B) Schematic of assay dose-response and interpretation. Fraction of cells fused (black curve) relative to cell count (toxicity measure, blue curve), both normalized by plate negative control, indicate compound efficacy (pink, strong inhibitor; gray, weak inhibitor; white, no-effect). Compounds are designated as effective if the maximum dose z-score is <-3. Strong vs. weak inhibitor designation is based on an arbitrary cutoff. (C) A panel of spike protease inhibitors (n = 10), which includes antagonists of both established SARS-CoV-2 entry pathways (cathepsin-dependent endocytosis vs. TMPRSS2/furin-mediated direct fusion), was tested. Mean and SEM: n = 4 biological replicates (16 images per). (D) Similar to (C), but displaying dose-response relationships for select inhibitors of indicated routes of endocytosis (e.g. clathrin, macropinocytosis) and steps in secretory pathway (e.g. ER-Golgi transport). See *Figure 3—figure supplement 1F* for additional tested compounds. (E) Similar to (C), but displaying dose-response relationships for compounds that strongly inhibit cell-cell fusion in ACE2-U2OS heterokaryon assay. See *Figure 3—figure supplement 2* for similar effect in VeroE6 assay (no exogenous ACE2 expression). (F) Summary of targeted drugs (n = 41) in U2OS and Vero based co-culture assays. Identified inhibitors are largely similar between cell types. Cell type-specific effects are noted. See also *Figure 3—figure supplements 1–2*; *Supplementary file 4*.

The online version of this article includes the following figure supplement(s) for figure 3:

**Figure supplement 1.** A novel high-throughput screening platform identifies modulators of syncytia formation.
**Figure supplement 2.** A novel high-throughput screening platform identifies modulators of syncytia formation.

performed for 1–6 hr: fusion was detectable by 1 hr, and Z'-factor peaked by 4–5 hr (*Figure 3—figure supplement 1B*).

Armed with a tractable assay (*Figure 3A,B*), we sought to characterize essential pathways for fusion, employing dose-response studies of a panel of drugs with well-characterized mechanism of action. Given that protease-mediated S2' cleavage is essential for cell-cell fusion (*Figure 1—figure supplement 1C*), we anticipated a large effect for specific classes of protease inhibitors. We thus tested a panel of spike protease inhibitors (n = 10), which included antagonists of both established SARS-CoV-2 entry pathways (cathepsin-mediated endocytic vs. TMPRSS2/furin-dependent direct fusion) (*Hoffmann et al., 2020a*; *Hoffmann et al., 2020b*; *Hoffmann et al., 2020c*; *Kawase et al., 2012*; *Millet and Whittaker, 2014*; *Ou et al., 2020*; *Shirato et al., 2018*; *Zhou et al., 2015*). Surprisingly, the antiretroviral protease inhibitor nelfinavir was unique in blocking fusion (*Figure 3C*), a compound whose therapeutic potential was identified by others (*Musarrat et al., 2020*). Given that other serine protease inhibitors (AEBSF, leupeptin, camostat) lacked efficacy, inhibition by nelfinavir may be related to its proteolysis-independent targets (*Brüning et al., 2013*; *De Gassart et al., 2016*; *Kirby et al., 2011*).

Next, we screened drugs that target specific routes of endocytosis or steps in the secretory pathway. Notably, no single endocytic route (clathrin, caveolae, macropinocytosis) was essential (*Figure 3D*; *Figure 3—figure supplement 1E,F*), and markers of each pathway did not co-localize with spike at sites of membrane-fusion or in the vesicles that resulted (*Figure 3—figure supplement 1A*). Nevertheless, certain drugs prevented syncytia formation (*Figure 3E,F*). For example, apilimod, a promising COVID-19 drug candidate that inhibits PIKFYVE kinase (*Cai et al., 2013*; *Kang et al., 2020*; *Riva et al., 2020*), was particularly potent (*Figure 3E*), however less so than in the case of infection studies (*Kang et al., 2020*; *Riva et al., 2020*). Inhibition of actin polymerization blocked fusion (*Figure 3E*), consistent with a putative role for cortical actomyosin-generated mechanical forces in fusion pore formation (*Figure 1*; *Chan et al., 2020b*; *Shilagardi et al., 2013*). Finally, several drugs that perturb membrane lipid composition were identified (MBCD/lovastatin, genistein, nystatin) (*Figure 3E*). These compounds could conceivably act by permeabilizing the plasma membrane. To rule out this possibility, cell fusion was examined following treatment with membrane permeabilizers digitonin or ethanol, both of which had no effect on fusion at non-toxic concentrations (*Figure 3—figure supplement 1C*). Finally, most compounds acted similarly in the VeroE6 heterokaryon assay (*Figure 3F*; *Figure 3—figure supplement 2*), which expresses endogenous ACE2. We therefore conclude that our syncytium-based screening platform can identify fusion-inhibiting drugs that act independently of canonical entry pathways and might uncover yet-to-be described determinants of membrane fusion.

## A drug repurposing screen implicates membrane lipid composition in cell-cell fusion

To gain further mechanistic insight into cell-cell fusion, we performed a drug repurposing screen of ~6000 small molecules at 30 μM (*Figure 4A*). Of these, 167 (2.8%) were inhibitory 'hits', which signified non-toxic compounds with a decrease in fusion greater than 3-standard deviations from the mean (z-score<-3) (*Figure 4B*; *Supplementary file 1*). To validate, we performed 7-point dose-responses for the top-80 most potent compounds, the vast majority of which replicated (*Figure 4C*; *Figure 4—figure supplement 1A,B*). Compounds were then unblinded to select experimenters. Reassuringly, 23 of these hits were redundant across the combined libraries (*Figure 4—figure supplement 1B*), and several were identified in previous virus entry screens (*Caly et al., 2020*; *Carbajo-Lozoya et al., 2012*; *Hoffmann et al., 2020c*; *Kindrachuk et al., 2015*; *Riva et al., 2020*; *Yamamoto et al., 2016*). We eliminated batch-dependent effects, purchasing top compounds from independent vendors, and replicating dose-response measurements for 23 of 24 (*Figure 4—figure supplement 1A*). To assess cell type specificity, dose-response studies for the same molecules were performed in the VeroE6 heterokaryon assay (*Figure 4—figure supplement 1A*). In almost all cases, inhibition of fusion occurred at lower compound concentrations relative to the U2OS assay, possibly due to differences in ACE2 levels between cell lines (e.g. 1–5 μM exogenous ACE2 in U2OS cells is likely much higher than endogenous ACE2 in VeroE6 cells; see Materials and methods).

Given their unusually high $EC_{50}$ (>10 μM) (*Figure 4—figure supplement 1A*) and the rapid kinetics of fusion, identified small molecules might act directly on the lipid bilayer (*Tsuchiya, 2015*), possibly by virtue of shared physicochemical or structural features. To assess this, we compared 20 physicochemical parameters (ChemAxon) for non-hits vs. hits, using GPCR inhibitors (~35% of FDA-approved drugs) (*Sriram and Insel, 2018*) as a control library (*Figure 4D*; *Figure 4—figure supplement 2A,B*). Among several statistically significant differences, hits were more lipophilic (LogD) and featured a greater number of ring systems (*Figure 4D*). Reassuringly, little correlation was observed between $EC_{50}$ values and lipophilicity (*Figure 4E*), indicating that the trend is not a result of a general increase in lipophilicity with avidity, as is commonly observed for promiscuous compounds in phenotypic screens (*Tarcsay and Keserű, 2013*).

Next, we asked whether specific chemical scaffolds are over-represented in hit compounds relative to ineffective compounds (*Figure 4F*; *Figure 4—figure supplement 2C*). Two scaffold classes (and corresponding substructures) reached particularly high statistical enrichment: dicholorophe-nethyl-imidazoles (found in azole antifungals) and tetrahydropyran-containing macrocyclic lactones (found in both ivermectin- and rapamycin-like compounds) (*Figure 4C,F*; *Figure 4—figure supplement 2C*). Such molecules can directly interact with the plasma membrane (*François et al., 2009*), perturbing cholesterol (e.g. production, transport) (*Bauer et al., 2018*; *Mast et al., 2013*; *Trinh et al., 2017*; *Xu et al., 2010*), and have been implicated as promising repurposed drugs for COVID-19 treatment, albeit by different mechanism of action (*Caly et al., 2020*; *Gordon et al., 2020*; *Kindrachuk et al., 2015*; *Rajter et al., 2021*).

## Highly unusual membrane-proximal regions of spike are needed for fusion

Based on the prevalence of lipophilic hits from the small-molecule screen, we posited that membrane-proximal regions of spike and/or ACE2 associate with essential plasma membrane lipids (e.g. cholesterol) to facilitate cell-cell fusion. To test this, we replaced the transmembrane and cytoplasmic domains of both ACE2 and spike with the previously used B7 TM (*Figure 1B*, *Supplementary file 4*). While 'chimeric' ACE2 similarly promoted cell fusion relative to wild-type (WT), chimeric spike protein lost this ability (*Figure 5A*). To determine critical elements that differentiate WT and chimeric spike from one another, we mutated its transmembrane (TM) and cytoplasmic domains (*Figure 5B*), assessing fusion in co-culture models (*Figures 1A* and *3A*). Replacement of spike's transmembrane domain with single-pass TMs of unrelated proteins (B7, ITGA1) blocked fusion, despite similar subcellular localization and ACE2-binding (*Figure 5C,L*; *Figure 5—figure supplement 1A–C*). Inclusion of a small extracellular motif of B7 not only eliminated fusion, but also impaired the ability of the chimeric spike to form synapse-like clusters with ACE2 (*Figure 5A*). This is likely indicative of an essential role of spike's membrane-proximal aromatic residues in cholesterol engagement (*Hu et al., 2019a*), as suggested by work on related coronaviruses (*Corver et al.,*

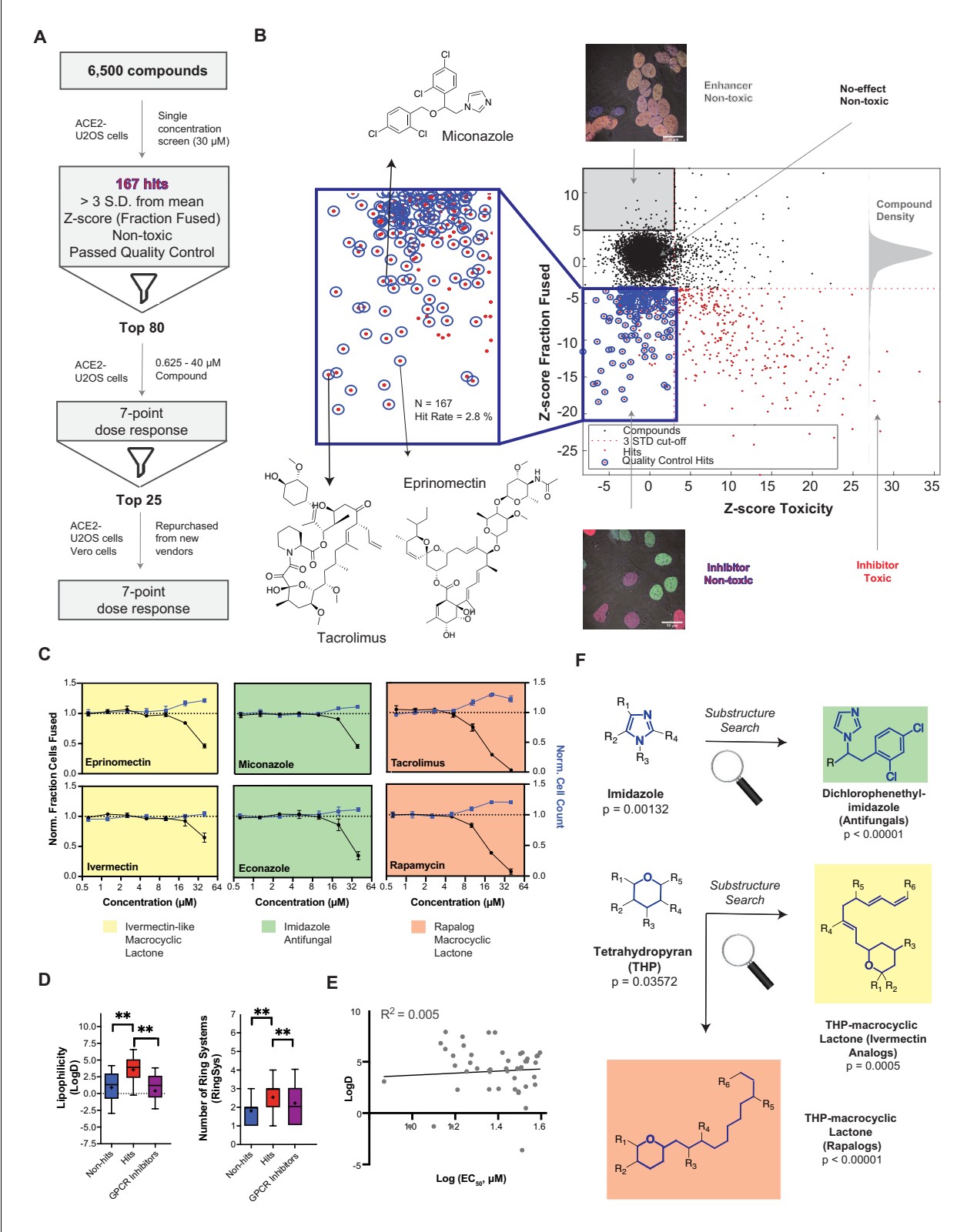

**Figure 4.** A drug repurposing screen implicates membrane lipid composition in cell-cell fusion. (**A**) High-throughput pipeline and workflow of small molecule screen (~6000 compounds, 30 µM) in ACE2-U2OS heterokaryon assay. 'Hits' refer to non-toxic compounds with a decrease in fusion of >3 standard deviations relative to plate negative control. 7-point dose-response was determined for top-80 inhibitors, followed by validation of select compounds (n = 24; obtained from different vendors) in both ACE2-U2OS and VeroE6 heterokaryon assays. (**B**) Results of compound screen. Plot:

*Figure 4 continued on next page*

*Figure 4 continued*

fraction fused vs. toxicity z-score. Red dots indicate compounds with decreased fusion (z-score<-3); blue inset, potential hits following toxicity filtering (z-score <3); blue circles, quality-controlled hits (inhibitory, non-toxic compounds with normal fluorescence); gray inset, compounds with increase in fraction cells fused (z-score >5) but no toxicity ('enhancers', see *Figure 6J*); right histogram, compound density as function of fraction fused z-score. Chemical structures are displayed for select validated hits. See *Supplementary file 1* for raw data. (C) Dose-responses for select hits in enriched substructure classes (see F): imidazoles (e.g. azole antifungals, green) and macrocyclic lactones (ivermectin-like, yellow; rapalogs, pink). Mean and SEM: n = 3 biological replicates (16 images per). (D) Box-and-whisker plots of select physicochemical properties (lipophilicity, logD; ring systems) for non-hits (blue), inhibitory hits (red), and GPCR inhibitors (purple) as calculated in ChemAxon. Boxes encompass 25–75% of variance; whiskers, 10–90%. Mean values are indicated by '+'; median values, lines. Statistical significance was assessed by Mann-Whitney U test: p-values of <0.05 and<0.001 are represented by * and **, respectively. (E) Lack of correlation between inhibitory hit $EC_{50}$ (see *Figure 4—figure supplement 1A*) and lipophilicity according to linear regression analysis conducted in GraphPad Prism. (F) Three substructure classes based on two scaffolds were identified to have high statistical enrichment in hits over non-hits: dicholorophenethyl-imidazoles (found in azole antifungals, green) and tetrahydropyrans with alkyl moieties (found in macrocyclic lactones; yellow and pink indicate ivemectin-like and rapalog compounds, respectively). See also *Figure 4—figure supplements 1–2*; *Supplementary file 1*.

The online version of this article includes the following figure supplement(s) for figure 4:

**Figure supplement 1.** A drug repurposing screen implicates membrane lipid composition in cell-cell fusion.

**Figure supplement 2.** A drug repurposing screen implicates membrane lipid composition in cell-cell fusion.

*2009*; *de Jesus and Allen, 2013*; *Epand et al., 2003*; *Liao et al., 2015*; *Lu et al., 2008b*; *Meher et al., 2019*).

In parallel, we serially truncated the spike cytoplasmic domain (CTD). Removal of its COPII-binding, ER-Golgi retrieval motif (*Cattin-Ortolá et al., 2020*; *McBride et al., 2007*) (1–1268) had no effect, nor did deletion of its subsequent acidic patch (1–1256) (*Figure 5C,L*; *Figure 5—figure supplement 1A–C*). However, removal of an additional 11 amino acids (1–1245) decreased fusion, and further truncation (1–1239) completely blocked it (*Figure 5C,L*; *Figure 5—figure supplement 1A–C*). Relative fusion correlated with overall cysteine content of the CTD (*Figure 5C*). These findings are consistent with previous studies on similar coronaviruses, which suggested that membrane-proximal cysteines are post-translationally modified with palmitoylated lipid moieties (*McBride and Machamer, 2010a*; *Petit et al., 2007*; *Sobocińska et al., 2017*).

Palmitoylated proteins typically feature only a few cysteines available for modification (*Chlanda et al., 2017*; *Wan et al., 2007*). We wondered whether spike CTD's peculiarly high cysteine content was unique amongst viral proteins, and performed a bioinformatic analysis of all viral transmembrane proteins, ranking them on maximal cysteine content in 20 amino-acid sliding windows (*Figure 5D–G*). Of all proteins in viruses that infect humans, SARS-CoV-2 spike features the highest cysteine content, followed closely by spike proteins in related coronaviruses, then hepatitis E ORF3 (*Figure 5G*; *Supplementary file 2*); it should be noted that ORF3 is palmitoylated and critical to viral egress (*Ding et al., 2017*; *Gouttenoire et al., 2018*). Consistent with studies on similar coronavirus spike proteins (*Liao et al., 2006*; *McBride and Machamer, 2010a*; *Petit et al., 2007*), mutagenesis of all spike cysteines to alanine severely diminishes cell-cell fusion in both U2OS and Vero models (*Figure 5I–L*; *Figure 5—figure supplement 1B–C*). To examine the role of cysteine palmitoylation, we assessed fusion upon treatment with palmitoylation inhibitor, 2-bromopalmitate (2-BP) (*Martin, 2013*). The effect was modest in U2OS cells, but more pronounced in Vero cells, suggesting that cysteine palmitoylation is indeed likely central (*Figure 5K*). However, we note that the EC50 for 2-BP is typically 10–15 µM (*Zheng et al., 2013*), which is lower than our obtained values. One possibility for the discrepancy is that our co-cultures are performed at high density, and synapse formation is fast (time scale of minutes) relative to biochemical pathways that modify subcellular localization (e.g. post-translational palmitoylation). Given the relatively modest and cell type-dependent effect of 2-BP treatment, future work using biochemical approaches will be required to confirm the role of palmitoylation and the precise mechanism by which spike's aromatic-rich transmembrane domain associates with cholesterol to drive membrane fusion.

Inspired by the synapse-like structures observed in living cells (*Figure 1*), we asked whether SARS-CoV-2 spike features amino acid motifs similar to human proteins that drive similar assemblies. Of the thousands of human transmembrane proteins, only 15 were 'spike-like', featuring both high membrane-proximal cysteine and aromatic content (*Figure 5H*; *Supplementary file 3*). Remarkably, the top four are all critical to forming specific types of adhesion junctions: three to tricellular tight

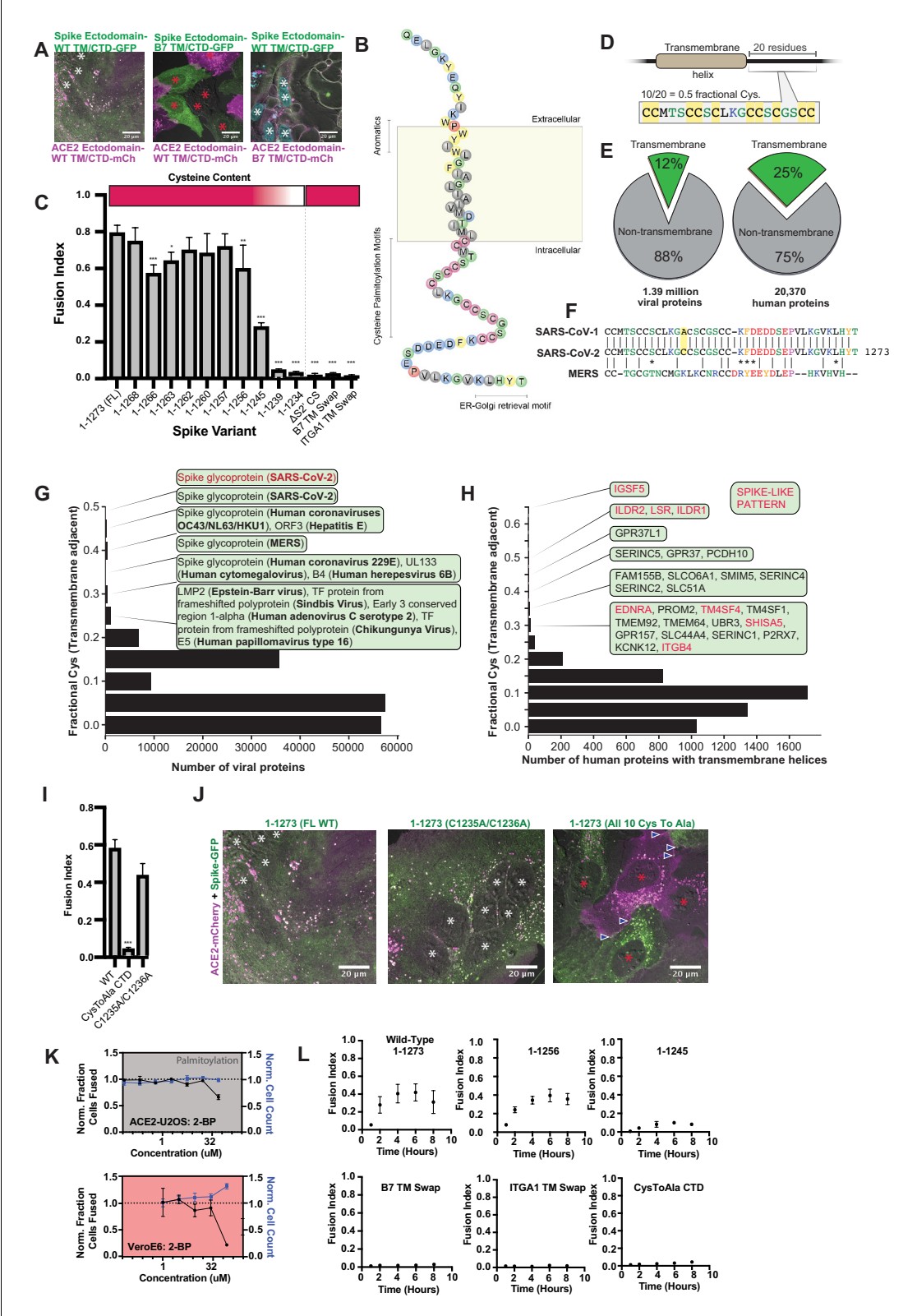

**Figure 5.** Highly unusual membrane-proximal regions of spike are needed for fusion. (**A**) Representative images of co-cultured (24 hr) U2OS cell lines, stably expressing indicated fluorescently tagged ACE2 or spike. 'B7 TM' indicates swap of endogenous transmembrane (TM) and cytoplasmic domain (CTD) of spike or ACE2 with that of the monomeric, single-pass, B7 transmembrane protein, along with its membrane-proximal extracellular region (30-amino acid spacer). White asterisks indicate nuclei in syncytia; red, in isolation. Note lack of arrowheads (synapses) in middle condition. See

*Figure 5 continued on next page*

Figure 5 continued

*Supplementary file 4* for residue composition of such 'chimeric' proteins. (B) Graphical representation of SARS-CoV-2 spike's TM alpha-helix and membrane-proximal regions, with residues colored by chemical properties (yellow, aromatic; cysteine, magenta; hydrophobic, gray; non-charged hydrophilic, green; charged hydrophilic; blue; proline, red). Of note: aromatic-rich region at ectodomain-TM interface, cysteine-rich cytoplasmic domain (CTD). (C) ACE2-U2OS heterokaryon assay but with co-cultured HNRNPA1-EYFP cells expressing spike variants (indicated). Relative CTD cysteine content for variants is depicted with heat map (top; dark red = more cysteines, white = none). Mean and SEM: n = 4 biological replicates (16 images per). See *Figure 5—figure supplement 1B* for all tested spike variants in both ACE2-U2OS and VeroE6 heterokaryon assays; *Figure 5—figure supplement 1C*, for representative images of ACE2-mCherry U2OS cells co-cultured with GFP-tagged spike variants. p-values of <0.01, <0.001, and <0.0001 are represented by *, **, and ***, respectively. (D) Cartoon representation of SARS-CoV-2 spike with highest cysteine content in a 20-amino acid sliding window, which guided bioinformatic analyses shown in (E–H). (E) Schematic of bioinformatic analysis performed, whereby 20-residue windows around the N- and C-terminal sides of transmembrane helices were scanned for local cysteine density. Pie charts: summary of total set of viral proteins retrieved and analyzed for the human virus proteome (left) and human proteome (right); green slice references proportion of proteins with one or more predicted transmembrane helixes. (F) Conservation of cysteine-rich CTD between spike proteins of highly pathogenic human coronaviruses. The only difference between the CTD of SARS-CoV-1 and SARS-CoV-2 is acquisition of an additional cysteine in the latter. MERS is substantially different, yet retains enrichment of cysteines. (G) Histogram of fractional cysteine scores for viral proteins, with high-fraction hits explicitly annotated. SARS-CoV-2 spike protein has the highest local cysteine density of any viral protein, closely followed by spike proteins from other coronaviruses. (H) Similar to (G), but for human proteins with one or more predicted transmembrane helix. Red: 'spike-like' transmembrane proteins with high cytoplasmic cysteine content and aromatic residues at ectodomain-membrane interface. (I) Similar to (C), but using spike variants with cysteines mutated to alanine (2 of 10 vs. 10 of 10). (J) Representative images for 24 hr co-culture of ACE2-mCherry (magenta) U2OS cells with those expressing GFP-tagged spike variant (green). White asterisks indicate nuclei in syncytia; red, those in isolation; arrowhead, synapses (select examples noted). (K) Dose-response relationship for 2-bromopalmitate (2-BP, inhibitor of cysteine palmitoylation), in both ACE2-U2OS (top) and VeroE6 (bottom) heterokaryon assays. Blue indicates number of nuclei (proxy for toxicity); black, percent cells fused; both normalized to DMSO control. Mean and SEM: n = 4 biological replicates (16 images per). (L) Similar to (C), but assesses kinetics of fusion by varying co-culture time prior to fixation. See *Figure 5—figure supplement 1A* for other tested spike variants. See also *Figure 5—figure supplement 1*; *Supplementary files 2–4*.

The online version of this article includes the following figure supplement(s) for figure 5:

**Figure supplement 1.** Highly unusual membrane-proximal regions of spike are needed for fusion.

junctions (ILDR1, ILDR2, LSR) and one to kidney/intestine tight junctions (IGSF5) (*Figure 5—figure supplement 1D*; *Higashi et al., 2013*; *Hirabayashi et al., 2003*). In light of the important role for palmitoylation in tricellular tight junction assembly (*Oda et al., 2020*), these findings suggest that SARS-CoV-2 may operate by a similar mechanism to promote adhesion and transcellular interfaces, an exciting possibility to be explored in future studies.

## Spike requires membrane cholesterol for fusion but via a raft-independent mechanism

Together, these data suggest that membrane fusion requires spike association with specific elements of the plasma membrane. If so, such assemblies would display slow dynamics relative to transmembrane proteins that more freely diffuse in the two-dimensional lipid bilayer. To test this idea, we utilized fluorescence recovery after photobleaching (FRAP) to determine the recovery rate of a fluorescent molecule in a bleached region, and thereby infer relative molecular diffusion coefficients (*Soumpasis, 1983*). FRAP experiments were performed on a series of GFP-tagged spike variants and controls (B7 TM and ACE2) to determine whether its transmembrane domain and/or cysteine-rich CTD influence diffusion. Recovery for GFP-tagged ACE2, B7 TM-anchored RBD, and the B7 transmembrane control were similar (*Figure 6C*), approximating diffusion times for commonly studied transmembrane proteins (*Day et al., 2012*). In contrast, RBD attached to the native TM/CTD of spike featured significantly reduced recovery, with FL spike displaying even slower dynamics (*Figure 6A–C*). Swapping the B7 TM for spike TM/CTD rescued the rapid recovery, whereas exchange of just the TM or removal of cysteine-containing regions had an intermediate effect (*Figure 6C*). Conversely, deletion of regions shown to bind specific intracellular proteins (e.g. COPII-binding ER-Golgi retrieval motif, 1268–1273) had no effect (*Figure 6C*), implicating lipid-protein and not protein-protein interactions in spike's dynamics.

Given that membrane-proximal regions (*Supplementary file 4*) of spike regulate diffusivity and fusogen behavior, an intriguing possibility is that such features conspire to facilitate engagement of cholesterol-rich membrane domains (or 'lipid rafts') (*Leventsal et al., 2020*; *Pelkmans and Helenius, 2003*; *Simons and Ikonen, 1997*). Our findings on the requirement for spike's cysteine residues in fusion is interesting in this context, since palmitoylation of other proteins can drive association with

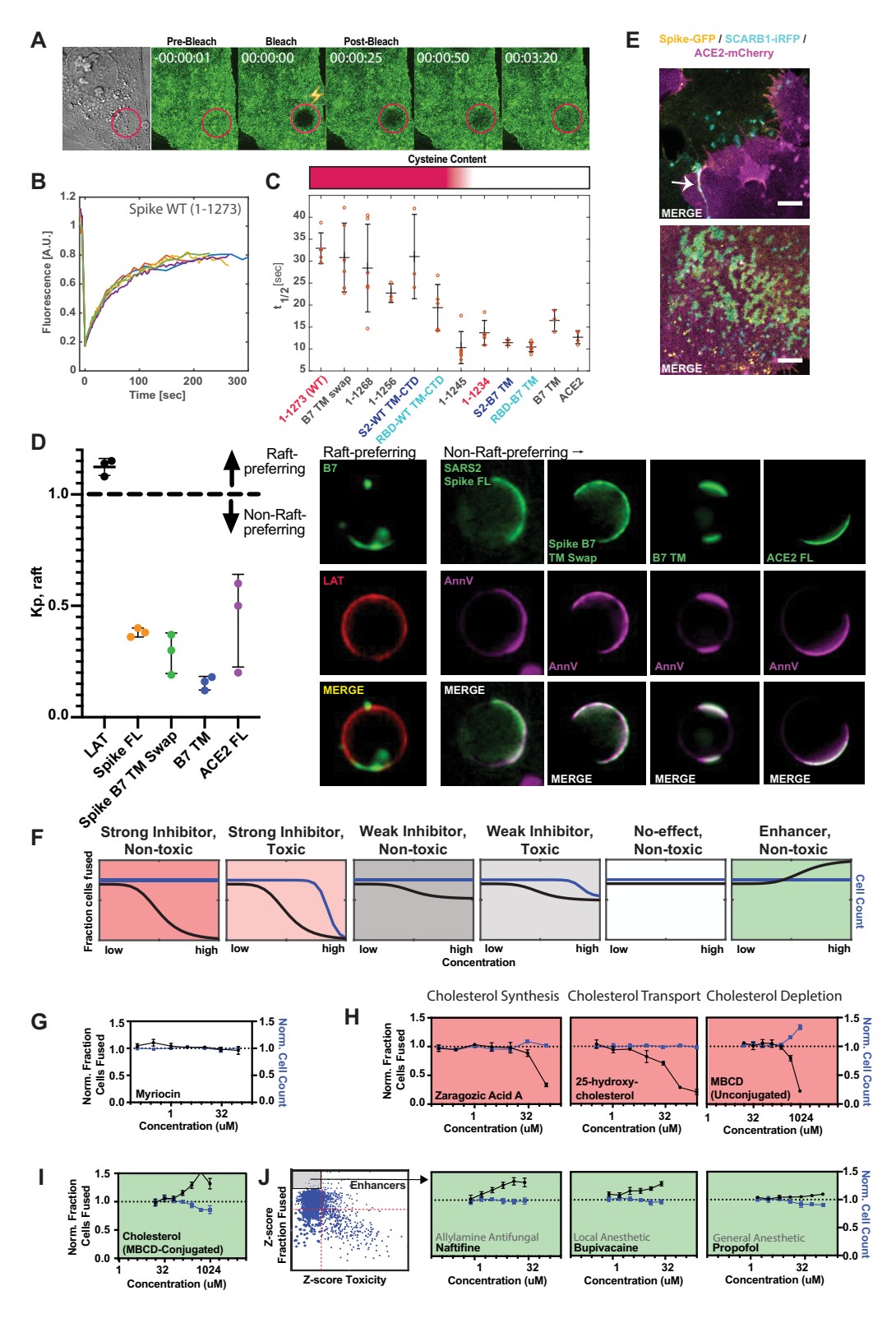

**Figure 6.** Spike requires membrane cholesterol for fusion but via a raft-independent mechanism. (**A**) Representative trial for fluorescence recovery after photobleaching (FRAP) of spike FL-GFP (green) on U2OS cell plasma membrane. Time since bleach (lightning bolt) of region of interest (red circle) is indicated. (**B**) Quantification of (**A**) and related trials, with each colored line specifying a separate FRAP experiment (n = 6 total). (**C**) Calculated half maximal fluorescence recovery ($t_{1/2}$) for indicated GFP-tagged spike variants. Each hollow red dot indicates the $t_{1/2}$ for a single FRAP trial. Mean and

*Figure 6 continued on next page*

*Figure 6 continued*

SEM: n = 4–6 technical replicates (one per cell). Heat map (top): relative cysteine content of tested spike variant (dark red = high cysteine content, white = none). (D) Ex vivo phase separation assay for relative partitioning ($K_p$) into lipid raft ordered phase ($L_o$) of giant plasma membrane vesicles (GPMVs). Left: quantification of GFP-tagged protein $K_p$ with DHPE serving as lipid raft/$L_o$ marker; AnnV, non-raft/$L_d$ protein marker; LAT, raft/$L_o$ protein marker. Mean and SEM: n = 3 biological replicates (colored dot;>10 GPMV technical replicates per). Right: representative images. (E) U2OS cells expressing spike-GFP (yellow) and SCARB1-iRFP (cyan) were co-cultured with ACE2-mCherry (magenta) cells and cell-cell fusion events were captured with live cell microscopy. Representative images show co-localization between spike and SCARB1 at synapses that precede fusion (top, arrow); and in extracellular deposits (bottom). Scale-bar, 5 μm. See *Figure 6—figure supplement 1A,B* for individual fluorescence channels. (F) Graphical schematic for ACE2-U2OS assay dose-response and interpretation. Fraction of cells fused (black curve) relative to cell count (blue curve), both normalized by the plate negative control, indicates compound effectiveness (pink, strong inhibitor; gray, weak inhibitor; white, no-effect; green, enhancer). (G) Lack of dose-dependent inhibition of fusion by sphingolipid-depleting, and raft-disrupting drug, myriocin, in ACE2-U2OS heterokaryon assay. Mean and SEM: n = 4 biological replicates (16 images per). (H) Similar to (G), but measuring effect of cholesterol-lowering drugs in ACE2-U2OS heterokaryon assay. Mean and SEM: n = 4 biological replicates (16 images per). (I) Similar to (G), but with MBCD-conjugated cholesterol, which increases plasma membrane cholesterol content. See *Figure 6—figure supplement 1E* for controls (i.e. other MBCD-conjugated lipids). (J) Similar to (G), but testing potential fusion-enhancing compounds (see *Figure 4B*, gray inset), which include allylamine antifungals (e.g. naftifine) and anesthetics (e.g. bupivacaine, propofol). See *Figure 6—figure supplement 1F* for similar effects by other small molecules belonging to these compound classes. See also *Figure 6—figure supplement 1*; *Supplementary file 4*.

The online version of this article includes the following figure supplement(s) for figure 6:

**Figure supplement 1.** Spike requires membrane cholesterol for fusion via a raft-independent mechanism.

these 10–50 nm protein-lipid clusters in the plasma membrane (*Levental et al., 2010*). While challenging to study in living cells, lipid rafts can be readily interrogated using ex vivo, phase separation assays as a micron-scale proxy for cholesterol association at the nano-scale (*Levental and Levental, 2015*; *Veatch, 2007*; *Veatch and Keller, 2003*). A particularly powerful example employs chemical-induced giant plasma membrane vesicles (GPMVs) from cells expressing a protein of interest (e.g. spike-GFP), allowing membrane components to reach equilibrium at reduced temperature (*Baumgart et al., 2007*; *Holowka and Baird, 1983*; *Levental et al., 2009*; *Sengupta et al., 2008*; *Veatch et al., 2008*). Given previous results using indirect detergent fractionation readouts (*McBride and Machamer, 2010b*; *Petit et al., 2007*), we were surprised to observe that SARS-CoV-2 spike does not partition strongly into GPMVs' dense, ordered phase ($L_o$) (*Figure 6D*), which is enriched for cholesterol and sphingolipids (*Levental et al., 2009*; *Levental et al., 2011*). Moreover, in our cell fusion assay, treatment with the raft-disrupting drug myriocin, which depletes sphingolipids from the plasma membrane (*Castello-Serrano et al., 2020*), did not inhibit fusion (*Figure 6F, G*). Thus, SARS-CoV-2 spike protein facilitates membrane-fusion in a manner that could be dependent on palmitoylation of its uniquely cysteine-rich CTD, but through a mechanism unique from canonical membrane nanodomains, although we cannot rule out a discrepancy in lipid raft properties between GPMVs and living cells (*Levental et al., 2020*).

Despite lack of partitioning into the cholesterol-rich ordered phase of GPMVs, we noted that long-term culture caused spike (but not ACE2) to accumulate in immobile deposits on the glass surface of the culture dish (*Figure 6E*; *Figure 6—figure supplement 1B*). Recent studies determined that cholesterol-rich membrane components are particularly prone to sloughing from the cell and sticking to glass surfaces (*He et al., 2018*; *Hu et al., 2019b*). We thus co-expressed the cholesterol-binding protein SCARB1 (*Linton et al., 2017*) with spike or stained spike cells with fluorescently labeled cholesterol. Both cholesterol markers co-localized with immobilized spike deposits (*Figure 6E*; *Figure 6—figure supplement 1A–C*). Taken together, the data suggests that spike potentially associates with a specific population of cholesterol, which is biochemically distinct from the sphingomyelin-associated lipid complexes enriched in canonical rafts (*Das et al., 2014*; *Endapally et al., 2019a*; *Kinnebrew et al., 2019*). Future studies will be needed to assess the nature of such cholesterol pools, potentially using toxin-based probes that discriminate between free and inaccessible forms (*Das et al., 2014*; *Endapally et al., 2019b*), and how each is affected by the identified lipophilic compounds (*Figure 4*).

To interrogate the role of cholesterol in cell fusion, we tested drugs that disrupt cholesterol synthesis (zaragozic acid) or reduce plasma membrane cholesterol (25-hydroxycholesterol; methyl-beta-cyclodextrin or 'MBCD') in the U2OS-ACE2 heterokaryon assay. All compounds inhibited fusion in a dose-dependent manner (*Figure 6H*). However, such drugs can indirectly lead to cholesterol-

independent changes in membrane lipid composition, especially at high concentrations (*Zidovetzki and Levitan, 2007*), and many require incubation periods longer than the duration of the cell-cell fusion assay to exert their full effect. To more directly study the role of cholesterol levels, we harnessed MBCD's ability to shuttle specific lipids into the plasma membrane (*Zidovetzki and Levitan, 2007*). Unlike MBCD-conjugated linoleic acid and oleic acid, cholesterol greatly enhanced fusion (*Figure 6I*; *Figure 6—figure supplement 1D,E*).

We surmised that the drug repurposing screen identified compounds that act similarly, thus implicating a counteracting plasma membrane property that increases fusion. Indeed, a small subset of compounds, which include allylamine antifungals (naftifine and terbinafine) and anesthetics (ropivacaine, bupivacaine, propofol), enhance fusion in a dose-dependent manner (*Figure 6J*; *Figure 6—figure supplement 1F*). Whether this is related to an opposing effect on lipid bilayer composition and dynamics relative to drugs that reduce fusion requires further inquiry using a suite of biophysical approaches. However, the latter possibility is intriguing in light of extensive literature on anesthetics and membrane mobility (*Cornell et al., 2017*; *Goldstein, 1984*; *Gray et al., 2013*; *Tsuchiya and Mizogami, 2013*).

## SARS-CoV-2 infection depends on membrane cholesterol of the virus but not the host cell

Our findings on ACE2/spike-mediated fusion, using both U2OS and VeroE6 cells, suggest that many effective compounds prevent fusion by depleting cholesterol from the plasma membrane (*Figure 6*). While the relevance of such drugs for syncytium formation and disease pathogenesis in vivo remains circumstantial (*Figure 2*), the data nonetheless has implications for virus assembly and entry. Specifically, we predict that such compounds would lack efficacy in virus entry models (*Chen et al., 2020*; *Dittmar et al., 2020*; *Riva et al., 2020*; *Wei et al., 2020*; *Zhu et al., 2020b*), instead requiring perturbation of the spike-containing virus membrane derived from the donor cell. To test this, we quantified spike-pseudotyped MLV particle entry into ACE2/TMPRSS2-expressing A549 acceptor cells (*Figure 7A*), which are primarily infected via the direct fusion pathway (*Hoffmann et al., 2020b*; *Shirato et al., 2018*; *Zhu et al., 2020b*). Apilimod, a PIKFYVE inhibitor and promising therapeutic in multiple SARS-CoV-2 models (*Kang et al., 2020*; *Riva et al., 2020*) including heterokaryon assays tested herein (*Figure 3E*; *Figure 3—figure supplement 2B*), inhibited (but did not completely block) entry at nanomolar concentrations (*Figure 7B*). This may indicate that multiple pathways (i.e. endocytic and direct fusion) are used to enter A549-ACE2/TMPRSS2 cells, which will be important to determine in future studies. By contrast, 25-hydroxycholesterol, which lowers plasma membrane cholesterol by redirection to the cell interior (*Abrams et al., 2020*; *Im et al., 2005*; *Wang et al., 2020*; *Yuan et al., 2020*; *Zang et al., 2020a*; *Zhu et al., 2020b*; *Zu et al., 2020*), had no effect (*Figure 7C*). However, MBCD, which directly 'strips' plasma membrane cholesterol without engaging intracellular targets (*Zidovetzki and Levitan, 2007*), blocked virus entry (*Figure 7D*).

Given the relative potency of MBCD in the pseudovirus entry model vs. syncytia assays (*Figure 6H*, *7D*), we hypothesized that it acts by depleting cholesterol from the viral membrane and not the host cell. To discriminate these possibilities, we used the same ACE2/TMPRSS2-expressing A549 cell line, but with a clinical isolate of SARS-CoV-2 virus (*Figure 7E*). Addition of virus in absence of drug resulted in significant infection of the cell layer, with many cells appearing as multi-nucleated following infection (*Figure 7F*). Pre-treatment of cells with millimolar doses of MBCD, which strongly inhibits both co-culture syncytia formation and pseudovirus entry, had no effect on infection as determined by RT-PCR and immunohistochemistry (*Figure 7F,G*). In striking contrast, pre-treatment of SARS-CoV-2 with low micromolar MBCD completely blocked infection (*Figure 7F,G*). Therefore, cholesterol content of SARS-CoV-2 viral particles, but not the host cell, is critical to infectivity (*Figure 7H*). However, the molecular basis of this cholesterol-dependent infectivity, that is whether it indeed results specifically from cholesterol-dependent spike fusogenicity or includes contributions from confounding effects such as large-scale virus permeabilization (*Graham et al., 2003*), remains to be determined.

## Discussion

Unprecedented resources devoted to the COVID-19 pandemic have allowed identification of promising stopgap therapeutic measures, as reaching effective vaccination coverage among populations

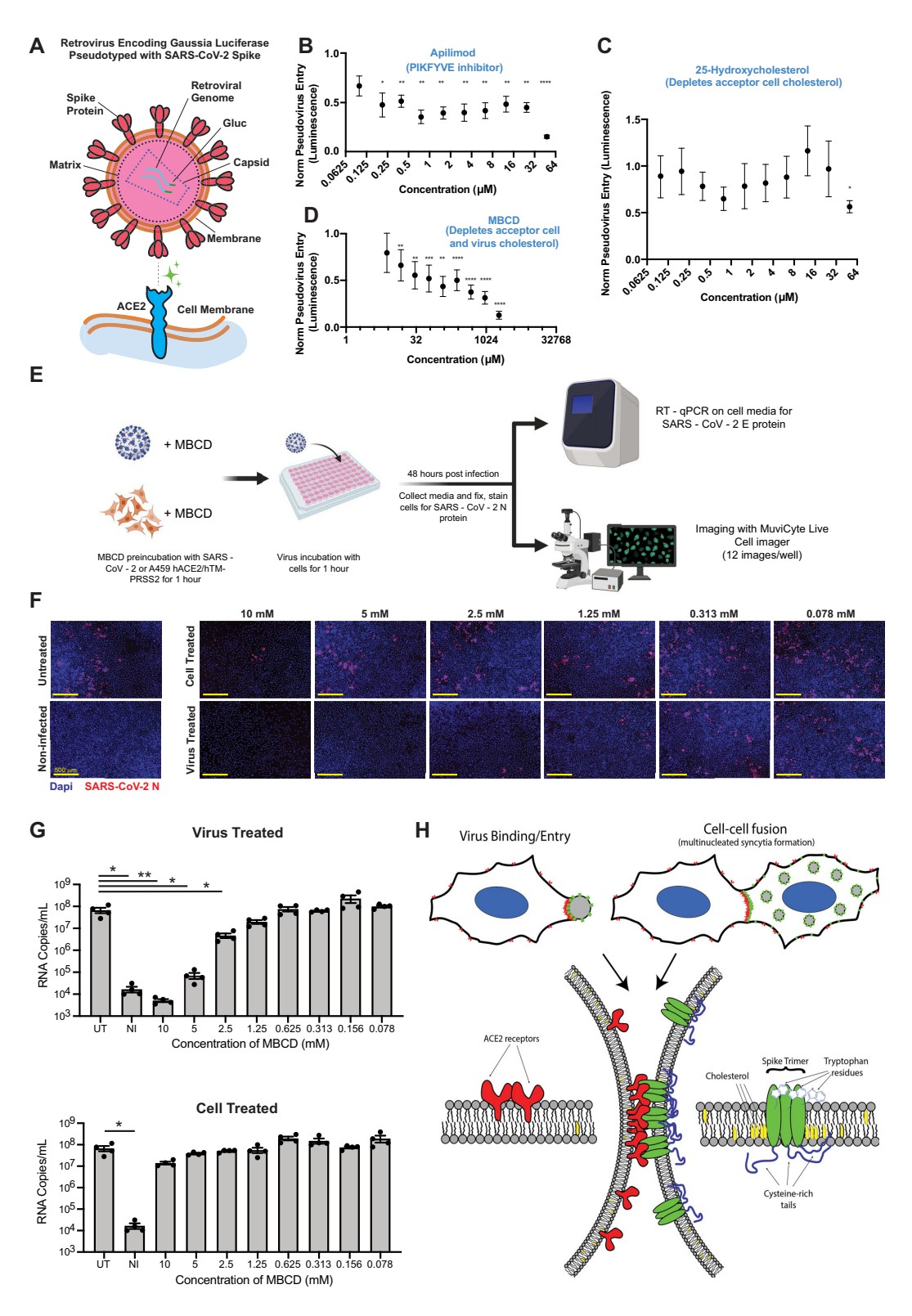

**Figure 7.** SARS-CoV-2 infection depends on membrane cholesterol of the virus but not the host cell. (**A**) Schematic representation of pseudotyped virus entry assay in ACE2/TMPRSS2-expressing A549 acceptor cells, which are primarily infected via direct fusion pathway. Pseudovirus encodes *Gaussia* luciferase gene, which allows luminescence-based measure of relative entry as a function of compound concentration. (**B**) Dose-dependent inhibition of pseudovirus entry (luminescence, arbitrary units) for a positive control compound (apilimod, PIKFYVE inhibitor), relative to control (1 = no effect;

*Figure 7 continued on next page*

*Figure 7 continued*

0 = complete block). Mean and SEM indicated for six replicates. P-values of <0.05, <0.01, <0.001, and <0.0001 are represented by *, **, *** and ****, respectively. (C) Similar to (B), but for cholesterol-transport disrupting drug, 25-hydroxycholesterol. (D) Similar to (B), but for plasma membrane cholesterol-stripping compound, MBCD. (E) Schematic of SARS-CoV-2 infection assays in ACE2/TMPRSS2 A549 acceptor cells. Relative infection is determined by RT-qPCR or immunohistochemistry of the SARS-CoV-2 nucleocapsid protein (N). (F) Representative immunofluorescence (nucleocapsid protein, red; nuclei/DAPI, blue) of A549 cells, 48 hr post-infection by SARS-CoV. Top: cells pre-treated with indicated dose of MBCD, followed by wash; bottom: pre-treatment of virus. (G) Similar to (F), but using RT-qPCR to quantify viral titer (RNA copies per mL cell media) following MBCD-treatment of virus (top) or cells (bottom). Identical controls plotted on both graphs for visualization purposes: UT = untreated cells, NI = non infected cells. Mean and SEM indicated for n = 4 independent biological replicates (black dots). p-values of <0.05 and<0.01 are represented by * and **, respectively. (H) Graphical model of the biomolecular interactions required for SARS-CoV-2 spike-mediated membrane fusion. Bottom: palmitoylated cysteines (blue) act as multivalent membrane contacts, anchoring trimeric spike peplomers (green) to the phospholipid bilayer (black) and potentially allowing transient higher order assemblies of trimers. Aromatic residues (e.g. tryptophans) at the spike ectodomain-membrane interface associate with accessible cholesterol (yellow) to promote synapse-like clusters with ACE2 receptors (red) on apposing membranes. Without these collective interactions, spike's fusion machinery (e.g. fusion peptide and heptad repeats) is unable to surmount the energetically costly barrier to lipid bilayer mixing, both in virus-cell (top, left) and cell-cell fusion (top, right).

will take time. Moreover, the global health impact of COVID-19 will linger for many years, since vaccination programs lag behind in developing nations, and vaccine-evading SARS-CoV-2 variants are constantly evolving (*Davies et al., 2021*; *Wibmer et al., 2021*). Virus entry-based assays were particularly critical to discovering essential receptors (ACE2) and proteases for SARS-CoV-2 infection, along with promising repurposed drugs (*Dittmar et al., 2020*; *Hoffmann et al., 2020a*; *Hoffmann et al., 2020b*; *Lan et al., 2020*; *Ou et al., 2020*; *Riva et al., 2020*; *Shang et al., 2020*; *Walls et al., 2020*; *Wei et al., 2020*; *Wrapp et al., 2020*; *Yan et al., 2020*; *Zhu et al., 2020b*). However, many fundamental aspects of the SARS-CoV-2 infectious cycle remain poorly understood, hampering efforts for effective treatment. In particular, commonly used approaches are poorly equipped to interrogate factors that contribute to the formation of a fusion-competent virus, particularly in a high-throughput manner amenable to both small molecule and genetic screens. Here, we show that in vitro SARS-CoV-2 spike and ACE2 cell co-culture assays overcome this limitation, and uncover a critical role for viral membrane composition in infection and formation of pathological syncytia.

Our approach relies on a combination of high-throughput screening, quantitative live cell imaging, and viral infection assays, all of which implicate biophysical aspects of the plasma membrane, particularly cholesterol-rich regions, in facilitating spike-mediated membrane fusion. While consistent with pioneering work on related coronaviruses (*Cervin and Anderson, 1991*; *Daya et al., 1988*) as well as more recent studies on SARS-CoV-2 (*Wang et al., 2020*; *Yuan et al., 2020*; *Zang et al., 2020a*; *Zhu et al., 2020b*; *Zu et al., 2020*), our approach allows unprecedented live cell access into the initial steps of membrane fusion. Further, our novel high-throughput cell-cell fusion assay not only negates safety concerns associated with a deadly virus, but may allow rapid interrogation of the pathogenic nature of newly emerging spike variants (*Davies et al., 2021*; *Wibmer et al., 2021*).

Cholesterol is known to preferentially interact with certain membrane proteins, particularly those modified with specific lipid moieties (e.g. palmitic acid) (*Levental et al., 2010*; *Martin, 2013*; *Sobocińska et al., 2017*), together clustering into nanodomains which have been referred to as lipid rafts (*Levental et al., 2020*; *Simons and Ikonen, 1997*; *Veatch and Keller, 2005*). In the context of SARS-CoV-2 spike, such cholesterol-rich nanodomains could potentially facilitate the energetically-unfavorable process of lipid bilayer mixing (*Heald-Sargent and Gallagher, 2012*; *Kim and Chen, 2019*; *Tenchov et al., 2006*). However, given its lack of partitioning into ordered GPMV domains (*Figure 6D*), spike may form nanoscale clusters by a different mechanism. A favored model is that an accessible population of cholesterol, independent of sphingolipids and rafts (*Das et al., 2014*; *Kinnebrew et al., 2019*), interacts directly with spike trimers and mediates formation of higher-order protein-lipid assemblies (*Figure 7H*). Precedent for this model is provided by raft-independent yet cholesterol-dependent mechanisms of biomolecular clustering essential for influenza infection (*Goronzy et al., 2018*; *Zawada et al., 2016*). Spike's cysteine-rich CTD could further amplify this effect, potentially via palmitoylated moieties, or, by directly promoting dynamic protein oligomers, similar to what was recently described for the *orthoreovirus* p14 FAST protein (*Chan et al., 2020a*). The interplay between oligomerization, palmitoylation, cholesterol association, and membrane dynamics, and how each of these properties are affected by compounds identified in our screen, will require additional methodologies beyond the scope of this study.

In strong support for a key role of lipid structure and composition in spike-mediated membrane fusion, many compound classes over-represented in our drug repurposing screen are implicated in perturbation of the lipid bilayer (e.g. antifungal azoles, rapalogs/mTOR inhibitors, ivermectin analogs) (*Figure 4C–F*; *Bauer et al., 2018*; *Head et al., 2017*; *Long et al., 2020*; *Mast et al., 2013*; *Trinh et al., 2017*; *Xu et al., 2010*). Interestingly, anti-fungals appear to be enriched in both fusion-inhibiting and -promoting hits. However, the latter tended to favor allylamines rather than azoles, implicating physicochemical and structural differences in direction of effect despite similar action on fungal ergosterol synthesis. We further note that our screen identified anesthetics as promoters of fusion (*Figure 4B*, *6J*). This is intriguing, given that anesthetics are chemically diverse, hydrophobic molecules, which perturb lipid mobility and ordering in membranes (*Cornell et al., 2017*; *Goldstein, 1984*; *Gray et al., 2013*; *Tsuchiya and Mizogami, 2013*). Whether such similarities can be extended to other fusion-promoting compounds remains to be determined, as is the root membrane-based property that discriminates them from drugs that prevent syncytia formation.

Our high-throughput screening approach uses transcellular membrane fusion as a proxy for virus entry into the host cell. However, beyond a screening tool, we present findings that suggest direct pathological relevance for syncytia. Specifically, human cell populations expressing ACE2 or spike cause cellular pathology with unexpected parallels to COVID-19 patient histology, who feature pervasive multinucleated cells in lung tissue (*Figure 2*; *Giacca et al., 2020*; *Tian et al., 2020*). These COVID-19 patient syncytia are similar to those found in other respiratory viral infections (*Frankel et al., 1996*; *Johnson et al., 2007*; *Maudgal and Missotten, 1978*). Whether COVID-19 syncytia arise by the cellular pathways described in this work is unclear, although recent data is supportive. For example, contemporary studies report syncytia in cultured cells following exposure to infectious SARS-CoV-2 (*Buchrieser et al., 2020*; *Cattin-Ortolá et al., 2020*; *Hoffmann et al., 2020a*; *Ou et al., 2020*; *Papa et al., 2020*; *Xia et al., 2020*; *Zang et al., 2020b*), an observation that we extend to ACE2/TMPRSS2 A549 cells (*Figure 7F*). Further, the synapses that precede fusion (*Figure 1*) superficially resemble the virus-filled filopodia observed following SARS-CoV-2 infection (*Bouhaddou et al., 2020*). We hypothesize that such structures arise via intercellular interactions between ACE2 and virus-unincorporated spike clusters, but cannot rule out the proposed kinase-based mechanism (*Bouhaddou et al., 2020*).

Whether or not syncytia play a major role in COVID-19 disease progression, the fact that lipid-targeting drugs disrupt spike-mediated membrane fusion has implications for treatment. Indeed, cholesterol-tuned viral infectivity was similarly shown using both spike-pseudotyped MLV and patient isolates of SARS-CoV-2. These data provide strong evidence that viral spike-cholesterol association, and not lipid composition of the host cell membrane or resultant endocytic pathway, mediates effects reported in this study (*Figure 7*) and perhaps others (*Zhu et al., 2020b*). Ultimately, viral membranes are produced from the membranes of infected host cells, suggesting that lipid-targeting treatments might disrupt formation of fusion-competent virus particles. Given the advanced state of hyperlipidemia drug development (e.g. statins) (*Goldstein and Brown, 2015*), insights from this work may be of immediate significance for COVID-19 treatment. Consistent with this, retrospective analyses of patient outcomes observed significant reductions in mortality for those prescribed cholesterol-lowering statins (*Daniels et al., 2020*; *Zhang et al., 2020*). A critical role for cholesterol in virus assembly may explain this observation, and could partially account for why COVID-19 risk factors (e.g. menopause, obesity, age) (*Costeira et al., 2020*; *Zhou et al., 2020*) similarly correlate with differential sterol processing. Additional work is clearly required, particularly with respect to the open question of what degree of cholesterol decrease would be required for effective treatment, but in the context of the rapidly evolving landscape of COVID-19 treatment options, our findings underscore the potential utility of statins and other lipid modifying treatments.

Invariably, opportunistic infections hijack physiological cellular processes to ensure their survival (*Pelkmans and Helenius, 2003*). To this end, we speculate that physiological and pathological synapses and resulting syncytia (or lack there of) arise in part from shared lipid bilayer properties at the nanoscale. Consistent with this concept, actin-dependent, ACE2/spike fusion events proceed from 'finger-like' projections and synapses between cells to fusion pore dilation and membrane collapse, closely resembling the orderly biogenesis of myoblast-derived syncytia (*Chen et al., 2008*; *Duan et al., 2018*; *Kim and Chen, 2019*; *Shi et al., 2017*; *Shilagardi et al., 2013*). Second, we highlight the remarkable similarities between SARS-CoV-2 spike and tricellular tight junction proteins (*Higashi et al., 2013*; *Oda et al., 2020*; *Sohet et al., 2015*), specifically with respect to membrane-

anchoring cysteines and aromatics (*Figure 5H*; *Figure 5—figure supplement 1D,E*). These observations suggest that both pathological viruses and adherent cells independently evolved proteins with abnormally strong affinity for the plasma membrane to ensure stability of transcellular complexes (*Figure 7H*), whether to initiate fusion or maintain tissue integrity. We thus envision that assays presented herein may have broad utility for understanding the biophysics of synapse and fusion pore assembly, representing an exciting example of how inquiry into viral pathogenesis illuminates physiological function.

## Materials availability

Plasmids and cell lines generated in this study are available from the lead contact.

## Experimental model and subject details

Select cell lines (VeroE6, Calu3, A549) were obtained from ATCC at the onset of the study and validated by the vendor. Following passage and usage by experimenters, all human cell lines (HEK293T, U2OS, Beas2B, Calu3, A549) were validated by STR profiling (ATCC) with 100% match between submitted samples and database profiles. All cell lines tested negative for mycoplasma (method: Universal Mycoplasma Detection Kit, ATCC 30–1012K). No commonly misidentified cell lines from the list maintained by the International Cell Line Authentication Committee were used for experiments in this study. Please see METHOD DETAILS for additional information on cell lines and culture conditions.

# Materials and methods

### Key resources table

| Reagent type (species) or resource | Designation | Source or reference | Identifiers | Additional information |
|---|---|---|---|---|
| Antibody | Mouse monoclonal anti-TTF-1 (clone 8G7G3/1) | Agilent Dako | IR05661-2 | (1:200) |
| Antibody | Rabbit polyclonal anti-SARS-CoV nucleocapsid (N) protein | Rockland | 200–401-A50 | (1:2000) |
| Antibody | Goat polyclonal anti-rabbit IgG- Alexafluor 568 | Thermo Fisher | A11011 | (1:1000) |
| Biological sample (virus) | 2019-nCoV (SARS-CoV-2)/ USA-WA1/2020 | Center for Disease Control, BEI | MN985325 | |
| Biological sample (*Homo sapiens*) | Formalin-fixed, paraffin-embedded, autopsy lung tissue (39 deceased patients or 'decedents') | Brigham and Women's Hospital, Autopsy Division | N/A | |
| Chemical compound, drug | Dulbecco's Modified Eagle Medium (DMEM), High Glucose, Pyruvate | Thermo Fisher Scientific | 11995065 | |
| Chemical compound, drug | Eagle's Minimum Essential Media (EMEM) | ATCC | ATCC 30–2003 | |
| Chemical compound, drug | Opti-MEM Reduced Serum Medium | Thermo Fisher | 31985062 | |
| Chemical compound, drug | Penicillin-Streptomycin (10,000 U/mL) | Thermo Fisher | 15140122 | |
| Chemical compound, drug | Fetal Bovine Serum, Inactivated | Atlanta Biologicals | S11150H | |
| Commercial assay or kit | In-Fusion HD Cloning Plus | Takara Bio | 638910 | |
| Chemical compound, drug | Lipofectamine 3000 Transfection Reagent | Thermo Fisher Scientific | L3000008 | |
| Chemical compound, drug | Polybrene | Sigma-Aldrich | TR-1003-G | |

*Continued on next page*

*Continued*

| Reagent type (species) or resource | Designation | Source or reference | Identifiers | Additional information |
|---|---|---|---|---|
| Chemical compound, drug | Paraformaldehyde (16%) | Electron Microscopy Services | 15710 | |
| Chemical compound, drug | Phusion High-Fidelity DNA Polymerase | New England Biolabs | M0530L | |
| Chemical compound, drug | Texas Red DHPE | Thermo Fisher | T1395MP | |
| Recombinant protein | Annexin V 647 | Thermo Fisher | A13204 | |
| Chemical compound, drug | Small molecule library: LOPAC-1280 (1278 compounds, 96-well) | Sigma-Aldrich | LO1280-1KT | |
| Chemical compound, drug | Small molecule library: LOPAC-Pfizer (90 Pfizer-drugs, 96-well) | Sigma-Aldrich | LO5100 | |
| Chemical compound, drug | Small molecule library: L1200 (355 kinase inhibitors, 96-well) | Selleck Chemicals | L1200 | |
| Chemical compound, drug | Small molecule library: L1900 (120 epigenetic compounds, 96-well) | Selleck Chemicals | L1900 | |
| Chemical compound, drug | Small molecule library: L2300 (378 cancer compounds, 96-well) | Selleck Chemicals | L2300 | |
| Chemical compound, drug | Small molecule library: L4200 (1364 FDA-approved drugs, 96-well) | TargetMol | L4200 | |
| Chemical compound, drug | Small molecule library: Spectrum collection (2400 bioactive compounds, 96-well) | Microsource Discovery | N/A | |
| Chemical compound, drug | Nelfinavir mesylate | R and D Systems | 3766/10 | |
| Chemical compound, drug | Heparin | Sigma-Aldrich | H3393\ | |
| Chemical compound, drug | Nocodazole | Sigma-Aldrich | 487928 | |
| Chemical compound, drug | Amiloride/EIPA | Sigma-Aldrich | A3085 | |
| Chemical compound, drug | Wortmannin | Sigma-Aldrich | W1628-1MG | |
| Chemical compound, drug | Chlorpromazine | Sigma-Aldrich | 215921–500 MG | |
| Chemical compound, drug | Filipin | Sigma-Aldrich | F4767-1MG | |
| Chemical compound, drug | Nystatin | Thermo Fisher | BP29495 | |
| Chemical compound, drug | Leupeptin | Sigma-Aldrich | L2884-5MG | |
| Chemical compound, drug | AEBSF | Sigma-Aldrich | A8456 | |
| Chemical compound, drug | Furin inhibitor II (polyarginine) | Sigma-Aldrich | 344931 | |
| Chemical compound, drug | Decanoyl-RVKR-CMK (chloromethylketone) | Tocris | 3501 | |
| Chemical compound, drug | Methyl-Beta-cyclodextrin (MBCD) | Sigma-Aldrich | C4555-1G | |
| Chemical compound, drug | Lovastatin | Sigma-Aldrich | 438185 | |
| Chemical compound, drug | Camostat mesylate | Sigma-Aldrich | SML0057 | |
| Chemical compound, drug | E-64D (EST) | Sigma-Aldrich | 33000 | |
| Chemical compound, drug | Ammonium chloride | Sigma-Aldrich | A9434 | |
| Chemical compound, drug | Chloroquine diphosphate | Sigma-Aldrich | C6628 | |
| Chemical compound, drug | Gefitinib | Sigma-Aldrich | SML1657 | |

*Continued on next page*

*Continued*

| Reagent type (species) or resource | Designation | Source or reference | Identifiers | Additional information |
|---|---|---|---|---|
| Chemical compound, drug | BAPTA-AM | Sigma-Aldrich | 196419 | |
| Chemical compound, drug | Latrunculin A | Invitrogen | 428026 | |
| Chemical compound, drug | Rottlerin | Sigma-Aldrich | 557370- | |
| Chemical compound, drug | Dynasore | Sigma-Aldrich | 324410 | |
| Chemical compound, drug | Sodium chlorate | Sigma-Aldrich | 403016 | |
| Chemical compound, drug | Protease inhibitor cocktail | Sigma-Aldrich | P1860-1ML | |
| Chemical compound, drug | Ouabain | Sigma-Aldrich | 4995–1 GM | |
| Chemical compound, drug | Rostafuroxin/PST2238 | Sigma-Aldrich | SML1139 | |
| Chemical compound, drug | Silmitasertib | Selleck Chemicals | S2248 | |
| Chemical compound, drug | R-406 | Selleck Chemicals | .S2194 | |
| Chemical compound, drug | Apilimod | Selleck Chemicals | S6414 | |
| Chemical compound, drug | Bafilomycin A | Enzo | BML-CM110 | |
| Chemical compound, drug | 2-Bromopalmitic acid (2 BP) | Sigma-Aldrich | 238422–10G | |
| Chemical compound, drug | MG132 | Sigma-Aldrich | M8699 | |
| Chemical compound, drug | Monensin | Sigma-Aldrich | M5273 | |
| Chemical compound, drug | Brefeldin A | Sigma-Aldrich | B5936 | |
| Chemical compound, drug | Cycloheximide | Sigma-Aldrich | C4859-1ML | |
| Chemical compound, drug | Dimethyl sulfoxide (DMSO) | Sigma-Aldrich | D2650 | |
| Chemical compound, drug | Actinomycin D | Sigma-Aldrich | A5156-1VL | |
| Chemical compound, drug | Staurosporine | AbCam | ab120056 | |
| Chemical compound, drug | ISRIB | Sigma-Aldrich | SML0843 | |
| Chemical compound, drug | 3-Methyladenine | Sigma-Aldrich | M9281 | |
| Chemical compound, drug | Z-FA-FMK | Selleck Chemicals | S7391 | |
| Chemical compound, drug | NSC 23766 | Selleck Chemicals | S8031 | |
| Chemical compound, drug | Z-VAD-FMK | Selleck Chemicals | S7023 | |
| Chemical compound, drug | AZD8055 | Selleck Chemicals | S1555 | |
| Chemical compound, drug | 25-Hydroxycholesterol | Sigma-Aldrich | H1015 | |
| Chemical compound, drug | Aprotinin (BPTI) | Selleck Chemicals | .S7377 | |
| Chemical compound, drug | Genistein | Selleck Chemicals | S1342 | |
| Chemical compound, drug | Oleic acid (MBCD-conjugated) | Sigma-Aldrich | O1257 | |
| Chemical compound, drug | Cholesterol (MBCD-conjugated) | Sigma-Aldrich | C4951 | |
| Chemical compound, drug | Linoleic acid (MBCD-conjugated) | Sigma-Aldrich | L5900 | |
| Chemical compound, drug | Myriocin | Sigma-Aldrich | M1177-5MG | |
| Chemical compound, drug | Zaragozic Acid A | Cayman | 17452 | |
| Chemical compound, drug | Digitonin | Sigma-Aldrich | D141 | |
| Chemical compound, drug | Bortezomib | Selleck Chemicals | S1013 | |
| Chemical compound, drug | Bupivacaine hydrochloride | Sigma-Aldrich | PHR1128 | |
| Chemical compound, drug | Ropivacaine hydrochloride | Selleck Chemicals | S4058 | |
| Chemical compound, drug | Naftifine hydrochloride | Cayman | 19234 | |
| Chemical compound, drug | Terbinafine hydrochloride | Cayman | 10011619 | |
| Chemical compound, drug | Propofol | Sigma-Aldrich | P-076–1 ML | |

*Continued on next page*

*Continued*

| Reagent type (species) or resource | Designation | Source or reference | Identifiers | Additional information |
|---|---|---|---|---|
| Chemical compound, drug | Puromycin | Sigma-Aldrich | P7255 | |
| Chemical compound, drug | Blasticidin S | VWR | 1859–25 | |
| Chemical compound, drug | DAPI | Sigma-Aldrich | D9542 | |
| Chemical compound, drug | Hoechst 33342 Solution | Thermo Fisher | 62249 | |
| Chemical compound, drug | AMG 9810 | Cayman | 14715 | |
| Chemical compound, drug | Apomorphine hydrochloride | ApexBio | B6936 | |
| Chemical compound, drug | Avanafil | Cayman | 23024 | |
| Chemical compound, drug | BML-277 | Cayman | 17552 | |
| Chemical compound, drug | Canagliflozin | Cayman | 11575 | |
| Chemical compound, drug | Chlorohexidine | Targetmol | T1147 | |
| Chemical compound, drug | CP-471474 | Cayman | 29442 | |
| Chemical compound, drug | Gossypol | ApexBio | N2135 | |
| Chemical compound, drug | Indacaterol maleate | ApexBio | B1369 | |
| Chemical compound, drug | JFD00244 | Cayman | 14648 | |
| Chemical compound, drug | Miconazole nitrate | Cayman | 15420 | |
| Chemical compound, drug | Moxidectin | ApexBio | B3611 | |
| Chemical compound, drug | PD-407824 | Cayman | 25989 | |
| Chemical compound, drug | Rapamycin | Targetmol | T1537 | |
| Chemical compound, drug | Ritanserin | Tocris | 1955 | |
| Chemical compound, drug | Rolapitant hydrochloride | Targetmol | T3724 | |
| Chemical compound, drug | Scriptaid | ApexBio | A4106 | |
| Chemical compound, drug | SKI II | Cayman | 10009222 | |
| Chemical compound, drug | Sorafenib | Cayman | 10009644 | |
| Chemical compound, drug | TAK-285 | MCE | HY-15196 | |
| Chemical compound, drug | Ticagrelor | Cayman | 15425 | |
| Chemical compound, drug | TW-37 | Cayman | 20999 | |
| Chemical compound, drug | UK-356618 | ApexBio | A4440 | |
| Chemical compound, drug | Vorapaxar | ApexBio | A8809 | |
| Chemical compound, drug | CholEsteryl BODIPY (FL C12) | Thermo Fisher | C3927MP | |
| Chemical compound, drug | Fibronectin from bovine plasma | Sigma-Aldrich | F1141-5MG | |
| Cell line (*Homo sapiens*, female) | HEK293T | Marc Diamond, UTSW | N/A | |
| Cell line (*H. sapiens*, female) | U2OS | Tom Muir, Princeton | N/A | |
| Cell line (*H. sapiens*, male) | Beas2B | Celeste Nelson, Princeton | N/A | |
| Cell line (*H. sapien, males*) | A549 (ACE2/TMPRSS2) | This study | N/A | |
| Cell line (*H. sapiens*, male) | Calu3 | ATCC | HTB-55 | |
| Cell line (*Cercopithecus aethiops*, female) | VeroE6 (monkey) | ATCC | CRL-1586 | |
| Cell line (*H. sapiens*, female) | U2OS, various | This study | N/A | |

*Continued on next page*

*Continued*

| Reagent type (species) or resource | Designation | Source or reference | Identifiers | Additional information |
|---|---|---|---|---|
| Cell line (*C. aethiops*, female) | VeroE6 (monkey), various | This study | N/A | |
| Recombinant DNA reagent | SCARB1-GFP | Addgene | #86979 | |
| Recombinant DNA reagent | GFP1-10 | Addgene | #80409 | |
| Recombinant DNA reagent | GFP11 × 7-mCherry-a-tubulin | Addgene | #70218 | |
| Recombinant DNA reagent | LAMP1-mCherry | Addgene | #45147 | |
| Recombinant DNA reagent | mCherry-CAV1 | Addgene | #55008 | |
| Recombinant DNA reagent | GFP-EEA1 | Addgene | #42307 | |
| Recombinant DNA reagent | SARS-CoV-1-Spike | Addgene | #145031 | |
| Recombinant DNA reagent | RAC1 (*H. sapiens*) in pANT7_cGST | DNASU | CD00632727 | |
| Recombinant DNA reagent | ACE2 cDNA in pcDNA3.1 | Genscript | OHu20260 | |
| Recombinant DNA reagent | SARS-CoV-2 (2019-nCoV) Spike ORF (Codon Optimized) | Sino Biological | VG40589-UT | |
| Recombinant DNA reagent | pCAGGS-SARS-CoV2-S | BEI Resources, NIH | N/A | |
| Recombinant DNA reagent | pMLV gag-pol | Ploss Lab, Princeton | N/A | |
| Recombinant DNA reagent | pLMN8-Gluc | This study | N/A | |
| Recombinant DNA reagent | p-LAT Transmembrane-mRFP | This study | N/A | |
| Recombinant DNA reagent | PSP (lentivirus packaging plasmid) | Marc Diamond, UTSW | N/A | |
| Recombinant DNA reagent | VSVG (lentivirus packaging plasmid) | Marc Diamond, UTSW | N/A | |
| Recombinant DNA reagent | FM5 lentiviral vector (Ubiquitin C promoter) | *Sanders et al., 2014* | N/A | |
| Recombinant DNA reagent | FM5-mGFP-Standardized glycine-serine (GS) Linker-AscI Site-STOP | *Sanders et al., 2020* | N/A | |
| Recombinant DNA reagent | FM5-mCherry-Standardized GS Linker-AscI Site-STOP | *Sanders et al., 2020* | N/A | |
| Recombinant DNA reagent | FM5-miRFP670-Standardized GS Linker-AscI Site-STOP | *Sanders et al., 2020* | N/A | |
| Recombinant DNA reagent | FM5-NheI Site-Standardized GS Linker-mGFP | *Sanders et al., 2020* | N/A | |
| Recombinant DNA reagent | FM5-NheI Site-Standardized GS Linker-mCherry | *Sanders et al., 2020* | N/A | |
| Recombinant DNA reagent | FM5-NheI Site-Standardized GS Linker-miRFP670 | *Sanders et al., 2020* | N/A | |
| Recombinant DNA reagent | FM5-NheI Site-Standardized GS Linker-EYFP | This study | N/A | |
| Recombinant DNA reagent | FM5-HNRNPA1-EYFP | This study | N/A | |
| Recombinant DNA reagent | FM5-FUS-mCherry | This study | N/A | |
| Recombinant DNA reagent | FM5-GFP1-10 | This study | N/A | |
| Recombinant DNA reagent | FM5-GFP11 × 7 | This study | N/A | |

*Continued on next page*

*Continued*

| Reagent type (species) or resource | Designation | Source or reference | Identifiers | Additional information |
|---|---|---|---|---|
| Recombinant DNA reagent | FM5-mGFP-EEA1 | This study | N/A | |
| Recombinant DNA reagent | FM5-miRFP670-EEA1 | This study | N/A | |
| Recombinant DNA reagent | FM5-mGFP-RAC1 | This study | N/A | |
| Recombinant DNA reagent | FM5-miRFP670-RAC1 | This study | N/A | |
| Recombinant DNA reagent | FM5-mGFP-CAV1 | This study | N/A | |
| Recombinant DNA reagent | FM5-miRFP670-CAV1 | This study | N/A | |
| Recombinant DNA reagent | FM5-LAMP1-mGFP | This study | N/A | |
| Recombinant DNA reagent | FM5-LAMP1- miRFP670 | This study | N/A | |
| Recombinant DNA reagent | FM5-SCARB1-mGFP | This study | N/A | |
| Recombinant DNA reagent | FM5-SCARB1-miRFP670 | This study | N/A | |
| Recombinant DNA reagent | FM5-MYMK-mGFP | This study | N/A | |
| Recombinant DNA reagent | FM5-MYMK-mCherry | This study | N/A | |
| Recombinant DNA reagent | FM5-MYMK-miRFP670 | This study | N/A | |
| Recombinant DNA reagent | FM5-MYMX-mGFP | This study | N/A | |
| Recombinant DNA reagent | FM5-MYMX-mCherry | This study | N/A | |
| Recombinant DNA reagent | FM5-MYMX-miRFP670 | This study | N/A | |
| Recombinant DNA reagent | FM5-p14 (*Reptilian orthoreovirus*)-mGFP | This study | N/A | |
| Recombinant DNA reagent | FM5-p14 (*Reptilian orthoreovirus*)-mCherry | This study | N/A | |
| Recombinant DNA reagent | FM5-p14 (*Reptilian orthoreovirus*)- miRFP670 | This study | N/A | |
| Recombinant DNA reagent | FM5-Albumin signal sequence (SS)—8 amino acid GS Linker-B7 Transmembrane (TM)-mGFP | This study | N/A | |
| Recombinant DNA reagent | FM5-Albumin SS-8 amino acid GS Linker-B7 TM-mCherry | This study | N/A | |
| Recombinant DNA reagent | FM5-Albumin SS-8 amino acid GS Linker-B7 TM-miRFP670 | This study | N/A | |
| Recombinant DNA reagent | FM5-ACE2-mGFP | This study | N/A | |
| Recombinant DNA reagent | FM5-ACE2-mCherry | This study | N/A | |
| Recombinant DNA reagent | FM5-ACE2-miRFP670 | This study | N/A | |
| Recombinant DNA reagent | FM5-ACE2 Ectodomain-8 amino acid GS Linker-B7 TM/CTD-mGFP | This study | N/A | |
| Recombinant DNA reagent | FM5-ACE2 Ectodomain-8 amino acid GS Linker-B7 TM-mCherry | This study | N/A | |
| Recombinant DNA reagent | FM5-ACE2 Ectodomain-8 amino acid GS Linker-B7 TM-miRFP670 | This study | N/A | |
| Recombinant DNA reagent | FM5-SARS-CoV-1 Spike-mGFP | This study | N/A | |
| Recombinant DNA reagent | FM5-SARS-CoV-1 Spike-mCherry | This study | N/A | |

*Continued on next page*

*Continued*

| Reagent type (species) or resource | Designation | Source or reference | Identifiers | Additional information |
|---|---|---|---|---|
| Recombinant DNA reagent | FM5-SARS-CoV-1 Spike-miRFP670 | This study | N/A | |
| Recombinant DNA reagent | FM5-Spike Full-length (FL) (1–1273)-mGFP | This study | N/A | |
| Recombinant DNA reagent | FM5-Spike Full-length (FL) (1–1273)-mCherry | This study | N/A | |
| Recombinant DNA reagent | FM5-Spike Full-length (FL) (1–1273)-miRFP670 | This study | N/A | |
| Recombinant DNA reagent | FM5-Spike SS (1-12)-RBD (319-541)—8 amino acid GS Linker-B7 TM/CTD-mGFP | This study | N/A | |
| Recombinant DNA reagent | FM5-Spike SS (1-12)-RBD (319-541)—8 amino acid GS Linker-B7 TM/CTD-mCherry | This study | N/A | |
| Recombinant DNA reagent | FM5-Spike SS (1-12)-RBD (319-541)—8 amino acid GS Linker-B7 TM/CTD-miRFP670 | This study | N/A | |
| Recombinant DNA reagent | FM5-Spike SS (1-12)-RBD (319-541)-WT TM/CTD (1203–1273)-mGFP | This study | N/A | |
| Recombinant DNA reagent | FM5-Spike SS (1-12)-RBD (319-541)-WT TM/CTD (1203–1273)-mCherry | This study | N/A | |
| Recombinant DNA reagent | FM5-Spike SS (1-12)-RBD (319-541)-WT TM/CTD (1203–1273)-miRFP670 | This study | N/A | |
| Recombinant DNA reagent | FM5-Spike Ectodomain (1–1213)—8 amino acid GS Linker-B7 TM/CTD-mGFP | This study | N/A | |
| Recombinant DNA reagent | FM5-Spike Ectodomain (1–1213)—8 amino acid GS Linker-B7 TM/CTD-mCherry | This study | N/A | |
| Recombinant DNA reagent | FM5-Spike Ectodomain (1–1213)—8 amino acid GS Linker-B7 TM/CTD-miRFP670 | This study | N/A | |
| Recombinant DNA reagent | FM5-Spike B7 TM Swap with Extracellular Linker (1-1213-8 amino acid GS linker-B7 TM-1235–1273)-mGFP | This study | N/A | |
| Recombinant DNA reagent | FM5-Spike B7 TM Swap with Extracellular Linker (1-1213-8 amino acid GS linker-B7 TM-1235–1273)-miRFP670 | This study | N/A | |
| Recombinant DNA reagent | FM5-Spike B7 TM Swap (1–1213-B7 TM-1235–1273)-mGFP | This study | N/A | |
| Recombinant DNA reagent | FM5-Spike B7 TM Swap (1–1213-B7 TM-1235–1273)-miRFP670 | This study | N/A | |
| Recombinant DNA reagent | FM5-Spike ITGA1 TM Swap (1–1213-ITGA1 TM-1235–1273)–mGFP | This study | N/A | |

*Continued on next page*

*Continued*

| Reagent type (species) or resource | Designation | Source or reference | Identifiers | Additional information |
|---|---|---|---|---|
| Recombinant DNA reagent | FM5-Spike ITGA1 TM Swap (1–1213-ITGA1 TM-1235–1273)-miRFP670 | This study | N/A | |
| Recombinant DNA reagent | FM5-Spike 1–1268-mGFP | This study | N/A | |
| Recombinant DNA reagent | FM5-Spike 1–1268-miRFP670 | This study | N/A | |
| Recombinant DNA reagent | FM5-Spike 1–1266-mGFP | This study | N/A | |
| Recombinant DNA reagent | FM5-Spike 1–1266-miRFP670 | This study | N/A | |
| Recombinant DNA reagent | FM5-Spike 1–1263-mGFP | This study | N/A | |
| Recombinant DNA reagent | FM5-Spike 1–1263-miRFP670 | This study | N/A | |
| Recombinant DNA reagent | FM5-Spike 1–1260-mGFP | This study | N/A | |
| Recombinant DNA reagent | FM5-Spike 1–1260-miRFP670 | This study | N/A | |
| Recombinant DNA reagent | FM5-Spike 1–1256-mGFP | This study | N/A | |
| Recombinant DNA reagent | FM5-Spike 1–1256-miRFP670 | This study | N/A | |
| Recombinant DNA reagent | FM5-Spike 1–1245-mGFP | This study | N/A | |
| Recombinant DNA reagent | FM5-Spike 1–1245-miRFP670 | This study | N/A | |
| Recombinant DNA reagent | FM5-Spike 1–1239-mGFP | This study | N/A | |
| Recombinant DNA reagent | FM5-Spike 1–1239-miRFP670 | This study | N/A | |
| Recombinant DNA reagent | FM5-Spike 1–1234 (ΔCTD)-mGFP | This study | N/A | |
| Recombinant DNA reagent | FM5-Spike 1–1234 (ΔCTD)-miRFP670 | This study | N/A | |
| Recombinant DNA reagent | FM5-Spike CysAlaMutant (C1235A/C1236A/C1240A/C1241A/C1243A/C1247A/C1248A/C1250A/C1253A/C1254A)-mGFP | This study | N/A | |
| Recombinant DNA reagent | FM5-Spike CysAlaMutant (C1235A/C1236A/C1240A/C1241A/C1243A/C1247A/C1248A/C1250A/C1253A/C1254A)-miRFP670 | This study | N/A | |
| Recombinant DNA reagent | FM5-Spike C1235A/C1236A-mGFP | This study | N/A | |
| Recombinant DNA reagent | FM5-Spike C1235A/C1236A-miRFP670 | This study | N/A | |
| Recombinant DNA reagent | FM5-Spike SS (1-12)-ΔNTD (319–1273)-mGFP | This study | N/A | |
| Recombinant DNA reagent | FM5-Spike SS (1-12)-ΔNTD (319–1273)-miRFP670 | This study | N/A | |
| Recombinant DNA reagent | FM5-Spike ΔRBD (1–318, 540–1273)-mGFP | This study | N/A | |
| Recombinant DNA reagent | FM5-Spike ΔRBD (1–318, 540–1273)-miRFP670 | This study | N/A | |
| Recombinant DNA reagent | FM5-Spike ΔFusionPeptide (1–787, 807–1273)-mGFP | This study | N/A | |

*Continued on next page*

*Continued*

| Reagent type (species) or resource | Designation | Source or reference | Identifiers | Additional information |
|---|---|---|---|---|
| Recombinant DNA reagent | FM5-Spike ΔFusionPeptide (1–787, 807–1273)-miRFP670 | This study | N/A | |
| Recombinant DNA reagent | FM5-Spike ΔHR1 (1–919, 971–1273)-mGFP | This study | N/A | |
| Recombinant DNA reagent | FM5-Spike ΔHR1 (1–919, 971–1273)-miRFP670 | This study | N/A | |
| Recombinant DNA reagent | FM5-Spike ΔHR2 (1–1162, 1203–1273)-mGFP | This study | N/A | |
| Recombinant DNA reagent | FM5-Spike ΔHR2 (1–1162, 1203–1273)-miRFP670 | This study | N/A | |
| Recombinant DNA reagent | FM5-Spike ΔBothCleavageSites ('CS') (Δ685/Δ686/Δ815/Δ816)-mGFP | This study | N/A | |
| Recombinant DNA reagent | FM5-Spike ΔBothCS (Δ685/Δ686/Δ815/Δ816)-miRFP670 | This study | N/A | |
| Recombinant DNA reagent | FM5-Spike ΔS1/S2CS (Δ685/Δ686)-mGFP | This study | N/A | |
| Recombinant DNA reagent | FM5-Spike ΔS1/S2CS (Δ685/Δ686)-miRFP670 | This study | N/A | |
| Recombinant DNA reagent | FM5-Spike ΔS2'CS (Δ815/Δ816)-mGFP | This study | N/A | |
| Recombinant DNA reagent | FM5-Spike ΔS2'CS (Δ815/Δ816)-miRFP670 | This study | N/A | |
| Recombinant DNA reagent | FM5-Spike S813G-mGFP | This study | N/A | |
| Recombinant DNA reagent | FM5-Spike S813G-miRFP670 | This study | N/A | |
| Recombinant DNA reagent | FM5-Spike S816G-mGFP | This study | N/A | |
| Recombinant DNA reagent | FM5-Spike S816G-miRFP670 | This study | N/A | |
| Recombinant DNA reagent | FM5-Spike R815A-mGFP | This study | N/A | |
| Recombinant DNA reagent | FM5-Spike R815A-miRFP670 | This study | N/A | |
| Recombinant DNA reagent | FM5-Spike R815K-mGFP | This study | N/A | |
| Recombinant DNA reagent | FM5-Spike R815K-miRFP670 | This study | N/A | |
| Recombinant DNA reagent | FM5-Spike R815A/S816G-mGFP | This study | N/A | |
| Recombinant DNA reagent | FM5-Spike R815A/S816G-miRFP670 | This study | N/A | |
| Recombinant DNA reagent | FM5-Spike R815K/S816G-mGFP | This study | N/A | |
| Recombinant DNA reagent | FM5-Spike R815K/S816G-miRFP670 | This study | N/A | |
| Recombinant DNA reagent | FM5-Spike S813G/R815A-mGFP | This study | N/A | |
| Recombinant DNA reagent | FM5-Spike S813G/R815A-miRFP670 | This study | N/A | |
| Recombinant DNA reagent | FM5-Spike S813G/R815K-mGFP | This study | N/A | |

*Continued*

| Reagent type (species) or resource | Designation | Source or reference | Identifiers | Additional information |
|---|---|---|---|---|
| Recombinant DNA reagent | FM5-Spike S813G/R815K-miRFP670 | This study | N/A | |
| Recombinant DNA reagent | FM5-Spike S1 (1–685)-WT TM/CTD (1203–1273)-mGFP | This study | N/A | |
| Recombinant DNA reagent | FM5-Spike S1 (1–685)-WT TM/CTD (1203–1273)-miRFP670 | This study | N/A | |
| Recombinant DNA reagent | FM5-Spike S1 (1–685)—8 amino acid GS Linker-B7 TM/CTD-mGFP | This study | N/A | |
| Recombinant DNA reagent | FM5-Spike S1 (1–685)—8 amino acid GS Linker-B7 TM/CTD-miRFP670 | This study | N/A | |
| Recombinant DNA reagent | FM5-Spike SS (1-12)-S2 (686–1273)-mGFP | This study | N/A | |
| Recombinant DNA reagent | FM5-Spike SS (1-12)-S2 (686–1273)-miRFP670 | This study | N/A | |
| Recombinant DNA reagent | FM5-Spike SS (1-12)-S2 (686–1213)—8 amino acid GS Linker-B7 TM/CTD Swap-mGFP | This study | N/A | |
| Recombinant DNA reagent | FM5-Spike SS (1-12)-S2 (686–1213)—8 amino acid GS Linker B7 TM/CTD Swap-miRFP670 | This study | N/A | |
| Software, algorithm | ChemAxon Physicochemical Property Analysis | Morgan et al., *Angew. Chem. Int. Ed.*, 2018 | https://chemaxon.com/ | |
| Software, algorithm | R Software Mann-Whitney U Statistical Analysis | Morgan et al., *Angew. Chem. Int. Ed.*, 2018 | https://www.r-project.org/ | |
| Software, algorithm | NCGC Scaffold Hopper | NCATS Chemical Genomics Centre (NCGC) | https://tripod.nih.gov | |
| Software, algorithm | RDKit Substructure Analysis | RDKit | https://www.rdkit.org/ | |
| Software, algorithm | Python Software | Python | https://www.python.org/ | |
| Software, algorithm | GraphPad Prism Nonlinear Regression | GraphPad | https://www.graphpad.com/ | |
| Software, algorithm | MATLAB R2017b | Mathworks | https://www.mathworks.com/ | |
| Software, algorithm | ImageJ | Schneider et al., *Nature Methods*, 2012 | https://imagej.nih.gov/ij/ | |
| Software, algorithm | PANTHER v14.0 Gene Ontology | Mi et al., *Nature Protocols*, 2019 | https://pantherdb.org | |
| Software, algorithm | BioRender (Paid Academic Plan) | BioRender | https://bio-render.com | |

## Plasmid construction

Lentiviral plasmids encoding fluorescently tagged proteins of interest were cloned as described in previous work (*Sanders et al., 2020*), which introduced monomeric fluorescent protein (mGFP, mCherry, miRFP670) lentiviral vectors (FM5, ubiquitin C promoter) with standardized linkers/overlaps to allow Gibson assembly-based cloning in high-throughput. With the exception of IDT-synthesized open-reading frames (p14, MYMK, MYMX), DNA fragments coding proteins of interest were

amplified by PCR (oligonucleotides synthesized by IDT; see KEY RESOURCES table for origin of cDNA template), using Phusion High-Fidelity DNA Polymerase (New England Biolabs or 'NEB'). Gibson assembly (In-Fusion HD cloning kit, Takara) was used to insert gel-purified DNA (Qiagen, Gel Extraction Kit) into the desired lentiviral vector, linearized by NheI restriction enzyme (NEB) or AscI restriction enzyme (NEB) digestion. DNA was extracted from transformed Stellar competent bacteria (Takara) by mini-prep (Qiagen). DNA inserts were confirmed by Sanger sequencing (GENEWIZ), reading from both ends of the open reading frame.

To generate pLMN8-Gluc, the secreted version of Gluc gene was PCR amplified using Q5 polymerase (NEB) using gene specific primers purchased from IDT. The PCR amplicon was inserted into the SfiI-digested pLMN8 plasmid (*Ploss et al., 2009*) via In-Fusion cloning (Takara). The sequence of the resulting pLMN8 plasmid encoding secreted Gluc (pLMN8-Gluc) was confirmed by Sanger sequencing (Eton Bioscience). Mammalian cell, codon optimized pCAGGS-SARS-CoV-2 spike (S) was kindly provided by BEI resources (NIH). Retroviruses pseudotyped with vesicular stomatitis virus G protein (VSV-G) were used as positive controls. pLMN8-Gluc and pMLV gag-pol plasmids were amplified in NEB stable competent cells (NEB); pCAGGS-SARS-CoV-2 spike and pVSV-G, DH5α competent cells (ThermoFisher), then plasmids were purified by Nucleobond Xtra Midi (Takara).

## Cell culture

A549, VeroE6, and Calu3 cells were obtained from the American Type Culture Collection (ATCC), and all human cell lines were validated by STR profiling (ATCC). 293 T cells were a kind gift from Marc Diamond (UT Southwestern); U2OS cells, Tom Muir (Princeton University); Beas2B cells, Celeste Nelson (Princeton University); A549 cells expressing human ACE2 and TMPRSS2 (A549-hACE2/TMPRSS2), Mohsan Saeed (Boston University). All cell lines were grown in Dulbecco's Modified Eagle's Medium (DMEM with high glucose and pyruvate, ThermoFisher) supplemented with 10% fetal bovine serum (FBS, Atlanta Biologicals) and 1% penicillin/streptomycin (P/S, Gibco), with the exception of: A549-ACE2/TMPRSS2, which were additionally maintained under puromycin (Sigma) and blasticidin (VWR) selection (both at 0.5 μg/mL); and Calu3 cells, which were grown in Eagle's minimum essential media (EMEM, ATCC) with 10% FBS and 1% P/S. All cells were propagated at 37˚ C in a 5% $CO_2$, 20% $O_2$ environment.

## Lentivirus production

Lentiviruses encoding fluorescently tagged proteins of interest were produced by using a previously optimized protocol (*Sanders et al., 2014*). Briefly, HEK293T cells were co-transfected with indicated FM5 construct and two helper plasmids (VSVG and PSP) with Lipofectamine-3000 (Invitrogen). Lipid-based transfection reagents were avoided with exception of virus production. Supernatant was collected 2–3 days post-transfection, cell debris was pelleted/discarded using centrifugation (1000xg), and lentivirus-containing media was used to infect indicated cell line in 96-well plates or stored at −80˚C.

## Generation of stable U2OS and VeroE6 cell lines

U2OS cells were selected for this study due to their flat morphology (ideal for live cell imaging), ease of lentivirus transduction, and absence of most proteins critical for SARS-CoV-2 fusion. For example, ACE2, TMPRSS proteases, and IFITM proteins are not detectable, whereas the FURIN enzyme is present at <500 copies per cell (*Beck et al., 2011*). Experimenters are thus able to control relative levels of such proteins in stable populations of U2OS cells by transducing different amounts of lentivirus. Approximate concentrations of exogenous fusion proteins are estimated in different subcellular compartments (e.g. plasma membrane, endoplasmic reticulum) using fluorescence correlation spectroscopy (FCS) calibration curves established for a given laser scanning confocal microscope (*Sanders et al., 2020*). In contrast to U2OS cells, VeroE6 cells express endogenous ACE2, but likewise, do not express TMPRSS2, and are infected by the endocytic pathway (*Hoffmann et al., 2020a*; *Hoffmann et al., 2020b*).

For both U2OS and VeroE6 cells, lentivirus transduction was performed in 96-well plates as described (*Sanders et al., 2020*) with minor modifications. For large, inefficiently packaged spike constructs, 180 μL lentivirus supernatant was added to a single well of 96-well dish prior to cell plating. For smaller constructs, 30 or 60 μL lentivirus supernatant was applied. Wells containing pre-

added lentivirus received PBS-washed and trypsinized cells, so cell populations were at ~10–20% confluency upon adhesion to the dish surface. Cells were grown for three days in lentivirus to obtain confluence, which maximizes viral transduction efficiency and protein expression (*Sanders et al., 2020*). Confluent cells, now stably expressing fusion proteins of interest, were washed with PBS, trypsinized, and passaged to fibronectin-coated glass (CellVis) for initial expression comparisons using confocal microscopy (Day 4). If expression was relatively low or all cells did not express the protein of interest, lentivirus transduction was repeated in 96-well dishes, up to two additional times. In the case of described spike variants, expression at the plasma membrane was similar, with the exception of select variants that suffered from misfolding problems and were not studied further (see below).

In parallel to preliminary expression check by confocal microscopy, stable cell populations were expanded first in 12-well then six-well dishes. At confluence in 6-well (~Day 10), cells were frozen in liquid nitrogen (freezing media = 90% fetal bovine serum, 10% DMSO) for long-term storage and subsequent use. The described multiple passage and expansion approach eliminates cells expressing lethal levels of fluorescent fusion proteins. Further, this protocol minimizes high over-expression arti-facts (e.g. stress response) and membrane perturbations common with lipid-based transfection reagents such as Lipofectamine-3000.

In all experiments, at least 90% (typically close to 100%) of cells in a given population expressed indicated protein(s) at time of freezing (~Day 10) and live cell imaging experiments (Day 10 or after). All stable cell populations featured <5 µM protein of interest on plasma membrane at time of live cell imaging experiments (much less if averaged across entire cell), as estimated by FCS calibration curves (*Sanders et al., 2020*). For ACE2, B7 TM, and Spike RBD cells, fusion protein was present at an estimated 1–5 µM at the plasma membrane. In the case of spike variants that acted similarly to the wild-type version, concentrations greater than 1 µM were unattainable, as toxicity occurred at higher levels. A negative correlation between spike variants' ability to promote fusion and maximal expression level was noted, likely because spike features a slight propensity to promote cell-cell fusion even in the absence of ACE2, perhaps due to weak interaction with a different receptor pro-tein. For example, SARS-CoV-1 spike could be expressed several fold higher than SARS-CoV-2 spike.

For all described fluorescent fusion proteins, proper localization to the cell's plasma membrane was confirmed by live cell confocal microscopy at ~Day 4 and ~Day 10 (relative to lentivirus transduc-tion). In the case of certain spike variants (e.g. Δ815/Δ816, 'ΔS2'CS'), expression was unexpectedly low despite multiple rounds of lentivirus transduction. In these cases, tagged protein was confined to the endoplasmic reticulum. Such behavior was independent of fluorescent tag (e.g. mCherry, miRFP670, mGFP) and is assumed to to indicate misfolding, which decreases the likelihood of neces-sary post-translational processing (e.g. glycosylation) for productive secretion. These constructs were discarded or not studied in detail, as relationship between phenotype and cleavage site (or domain deletion) would be impossible to determine. Due to misfolding of Δ815/Δ816 ("ΔS2'CS), additional S2' cleavage site variants (e.g. S813G) were generated to assess its role in cell-cell fusion. For these variants, localization to plasma membrane and subsequent binding to ACE2 was confirmed (transcel-lular synapses) and essentiality for membrane fusion was addressed.

## Live cell confocal microscopy

Stable cell lines were plated on fibronectin-coated, 96-well glass bottom dishes (Cellvis) and immedi-ately imaged (in the case of experiments requiring observation of individual fusion events) or follow-ing 24 hr culture (e.g. comparison of relative fusion between spike variants). A Nikon A1 laser-scanning confocal microscope equipped with 60x oil immersion lens (numerical aperture of 1.4) was used to collect confocal images. A humidified incubator kept cells at 37˚C and 5% CO$_2$. Proteins tagged with EYFP, mGFP ('GFP'), mCherry, or miRFP670 ('iRFP') were imaged with 488, 488, 560, and 640 nm laser lines, respectively, and settings were optimized to minimize photobleaching and to negate bleed-through between channels. With the exception of heterokaryon co-culture assays, all confocal microscopy was performed on living cells to eliminated fixation-associated artifacts in subcellular localization.

## MBCD TMR-cholesterol labeling and TIRF imaging

Cells were incubated with TopFluor TMR-cholesterol (Avanti Polar Lipids #810385) complexed with methyl-beta-cyclodextrin (MβCD) in a 1:10 ratio (chol: MβCD). Total Internal Reflection Fluorescence (TIRF) Microscopy Images were collected on Nikon A1R + STORM (Nikon Ti2 frame) equipped with 405 nm, 488 nm, 561 nm, and 640 nm laser sources (Nikon LUN-F), a Princeton Instruments ProEM EMCCD camera, and SR HP Apo TIRF 100x/1.49 oil lens (MRD01995). The N-STORM module was used in TIRF mode and the TIRF angle was adjusted manually.

## Heterokaryon co-culture assay

U2OS cells expressing SARS-CoV-2 spike-iRFP and ACE2-iRFP with their respective nuclear markers HNRNPA1-EYFP and FUS-mCherry were grown in 10-cm cell culture dishes (ThermoFisher), trypsinized with 0.05% EDTA-trypsin (ThermoFisher), resuspended in DMEM (10% FBS, 1% P/S), and mixed in 1:1 ratio. $5.4 \times 10^6$ cells were immediately seeded per well into a glass-bottomed 384-well plate (CellVis) to a total of 80 µL volume using a Multidrop Combi SMART liquid-handling dispenser (ThermoFisher). Unless indicated, cells were incubated at 37°C for 5-hr, fixed with 4% paraformaldehyde (Electron Microscopy Services 16% PFA stock solution from freshly opened glass ampules was added directly to media to minimize variability between wells) for 10-min, washed with DPBS (Gibco), and stained with Hoechst (200 ng/mL). For VeroE6 cells co-cultures, the above procedure was followed exactly, replacing the ACE2-expressing U2OS cells with VeroE6 cells expressing FUS-mCherry nuclear markers. Unlike U2OS cells (*Beck et al., 2011*), VeroE6 monkey cells feature endogenous ACE2 expression and are readily infected with SARS-CoV-2 virus (*Hoffmann et al., 2020a*; *Hoffmann et al., 2020b*). For all effective compounds, manual inspection of representative images was performed to rule out the possibility that small molecules acted by inhibiting nucleocytoplasmic transport: in this scenario, tagged RNA-binding proteins would accumulate in cytosol or be unevenly distributed between nuclei of a single syncytium. This was never observed. See Quantification and statistical analysis for details on statistical comparisons.

## Targeted compound dose-response assays

For the targeted compound screen (*Figure 3*), compounds were purchased and dissolved in water, methanol or DMSO to achieve stock solutions at ~2000-fold concentration commonly reported by literature. Serial dilutions (7-doses, threefold dilutions unless indicated) were prepared in 20 µL DMEM per well and $5.4 \times 10^6$ cells of each cell type (40 µL volume per) were added to a final volume of 100 µL (0.5% DMSO). Heterokaryon co-culture assays were carried out as described above. Compounds were determined to be effective if the maximum dose z-score was <-3. Strong vs. weak inhibitor designations were based on arbitrary cutoffs in relative fusion and were reproducible across independent experiments (i.e. dose-responses performed on separate days). See Quantification and statistical analysis for details on statistical comparisons.

## Unbiased drug repurposing screen

For drug repurposing screen of 5985 compound library (derived from seven different commercial small molecule libraries; see KEY RESOURCES table), the described ACE2-U2OS heterokaryon assay was carried out by adding co-culture to 384-well plates with compounds pre-dispensed. Specifically, 240 nL of compound (10 mM, in dissolved in DMSO) was added using ECHO 550 (Labcyte) liquid dispenser to generate a final compound concentration of 30 µM upon addition of 80 µL cell co-culture.

## Dose-response validation of hits from drug repurposing screen

For 7-point, dose-response assay, appropriate volumes of compound as 10 mM DMSO solution were dispensed using ECHO 550 (Labcyte) liquid dispenser to generate final concentrations of 40, 20, 10, 5.0, 2.5, 1.25, 0.625 µM upon addition of 80 µL cell co-culture. Wells were back-filled as necessary to keep the total DMSO volume of 320 nL consistent for all wells, including negative control, so as to maintain 0.4% DMSO concentration. To validate the top-24 hits, compounds were purchased from independent suppliers, dissolved in DMSO at 10 mM stock concentrations, and dispensed in 7-point dilutions according to procedure above. See Quantification and statistical analysis for details on statistical comparisons.

## Automated fixed cell confocal imaging and data acquisition

Heterokaryon assay development, characterization and high-throughput screening were carried out on a Eclipse Ti2 inverted scanning confocal microscope (Nikon) equipped with an automated Water Immersion Dispenser (WID). Wells were characterized by 16 full field of view regions (211 × 211 µm) imaged with a 60x, 1.2-numerical aperture, water-immersion, Nikon objective with 512 × 512 resolution. Bi-directional scanning with Hoechst (405 excitation/425–475 emission filter; channel 1), GFP (488/500–550; channel 2), and mCherry (561/570–620; channel 3) channels were acquired by a line series through a 50 µm pinhole at a rate of one image per second. An automated image acquisition protocol was developed in the Nikon NIS-Elements JOBS module to navigate within each well and over the 384-well plate. Automated image processing and all subsequent analyses were implemented in MATLAB R2017b.

## Fluorescence recovery after photobleaching (FRAP)

Stable U2OS cell lines expressing indicated GFP-labeled proteins of interest were cultured for 24 hr on a 96-well glass-bottom dish (CellVis) and imaged using a Nikon A1 laser-scanning confocal microscope as described. Photobleaching was performed by scanning a 488 nm laser over a circular region of interest ~6.5 µm in diameter, while focusing on the plasma membrane of single cells, validated by carefully tuning the focus to a plane bellow the non-fluorescent nuclei until they were no longer observable (compare transmitted light and fluorescent images in *Figure 6A*) and fluorescence signal within the surrounding area reached its maximum. See Quantification and statistical analysis for analysis.

## Protein partitioning measurements in giant plasma membrane vesicles (GPMVs)

Cell membranes were stained with 5 µg/ml of Texas Red DHPE or Annexin V 647 (ThermoFisher), respectively, red or far-red fluorescent lipid dyes that strongly partition to disordered phases (*Baumgart et al., 2007*; *Klymchenko and Kreder, 2014*; *Stone et al., 2017*). Following staining, GPMVs were isolated as described (*Sezgin et al., 2012*) from U2OS stable cells lines expressing the protein of interest (LAT results were obtained from transient co-transfections). Briefly, GPMV formation was induced by 2 mM N-ethylmaleimide (NEM) in hypotonic buffer containing 100 mM NaCl, 10 mM HEPES, and 2 mM $CaCl_2$, pH 7.4. To quantify partitioning, GPMVs were observed on an inverted epifluorescence microscope (Nikon) at 4°C after treatment with 200 µM DCA to stabilize phase separation; this treatment has been previously demonstrated not to affect raft affinity of various proteins (*Castello-Serrano et al., 2020*). The partition coefficient ($K_{p,raft}$) for each protein was calculated from fluorescence intensity of the construct in the raft and non-raft phase for >10 vesicles/trial, with multiple independent experiments (n = 3) for each construct.

## Generation of retroviral pseudoparticles

All pseudotyped retroviruses were generated by co-transfection of plasmids encoding (1) a provirus containing the Gaussia luciferase reporter gene (LMN8-Gluc), (2) mouse leukemia virus (MLV) gag-pol (*Ploss et al., 2009*), and (3) codon-optimized SARS-CoV-2 spike.

On the day prior to transfection, $1.4 \times 10^7$ 293 T cells were seeded in a 150 mm tissue culture dish. The following day, a total of 15 µg of total DNA was transfected using 90 µL X-tremeGENE HP DNA Transfection Regent (Roche). To generate luciferase reporter SARS CoV-2-Spp and VSV-Gpp controls, (1) pLMN8-Gluc, (2) MLV gag-pol, and (3) either SARS CoV-2 spike or VSV-G were co-transfected at a ratio of 4.5:4.5:1, giving rise to SARS-CoV-2pp and VSV-Gpp, respectively. No envelope pseudoparticles (NEpp) was also generated using (1) pLMN8-Gluc and (2) MLV gag-pol at a ratio of 1:1. Media was replaced after 6–18 hr with DMEM containing 3% FBS, nonessential amino acids (NEAA, 0.1 mM, ThermoFisher), HEPES (20 mM, ThermoFisher), polybrene (4 µg/mL, Sigma-Aldrich). Supernatants were harvested at 48- and 72 hr after transfection, pooled and filtered (0.45 µm pore size), aliquoted, and stored at −80°C until usage.

## Pseudovirus blocking assay

Blocking assays with luciferase reporter pseudovirus were performed in poly-L-lysine coated flat-bottom 96-well plates using $1.5 \times 10^4$ A549-ACE2-TMPRSS2 cells per well. The next day, all compounds

(10 mM diluted in DMSO) except for MBCD were diluted to 50 µM by DMEM containing 3% FBS, NEAA (0.1 mM), HEPES (20 mM), polybrene (4 µg/ml) and penicillin-streptomycin. MBCD (40 mM diluted in PBS) was diluted to 2 mM by DMEM containing 0.5% DMSO, 3% FBS, NEAA (0.1 mM), HEPES (20 mM), polybrene (4 µg/ml), and penicillin-streptomycin. The final concentration of DMSO for all compounds was 0.5%. Two-fold serial dilutions of all compounds were co-cultured with cells for 2 hr at 37°C, and subsequently, the same volume of pseudovirus was added into the cells and incubated for 4 hr at 37°C. After incubation, wells were washed once with 100 µl Hank's Buffered Saline Solution (HBSS, ThermoFisher), and the media changed to 100 µL DMEM containing 3% FBS, NEAA (0.1 mM), HEPES (20 mM), polybrene (4 µg/mL), and penicillin-streptomycin. Each plate featured positive (no pseudovirus) and negative (DMSO only) controls (n = 6 biological replicates) for quantification.

## Pseudovirus luciferase assay

Luciferase assay were performed 48 hr after incubation. The supernatants were collected to assess Gaussia luciferase activity using Genecopoeia Luc-Pair Renilla luciferase HS Assay Kit (GeneCopoeia) following the manufacturer's instruction and measured on a Tristar2 LB942 luminometer (Berthold Technologies). See Quantification and statistical analysis for details on statistical comparisons.

## SARS-CoV-2 isolate stock preparation and titration

All replication-competent SARS-CoV-2 experiments were performed in a biosafety level 3 laboratory (BSL-3) at the Boston University' National Emerging Infectious Diseases Laboratories. 2019-nCoV/USA-WA1/2020 isolate (NCBI accession number: MN985325) of SARS-CoV-2 was obtained from the Centers for Disease Control and Prevention and BEI Resources. To generate the passage 1 (P1) virus stock, Vero E6 cells, pre-seeded the day before at a density of 10 million cells, were infected in T175 flasks with the master stock, diluted in 10 mL final volume of Opti-MEM. Following virus adsorption to the cells at 37°C for 1 hr, 15 mL DMEM containing 10% FBS and 1x penicillin/streptomycin was added to the flask. The next day, media was removed, cell were rinsed with 1x PBS and 25 mL of fresh DMEM containing 2% FBS was added. Two days later, when the cytopathic effect of the virus was clearly visible, culture medium was collected, filtered through a 0.2 µm filter, and stored at −80°C. Our P2 working stock of the virus was prepared by infecting Vero E6 cells with the P1 stock, at a multiplicity of infection (MOI) of 0.1. Cell culture media was harvested at day 2 and day 3 post infection, and after the last harvest, ultracentrifuged (Beckman Coulter Optima L-100k; SW32 Ti rotor) for 2 hr at 25,000 rpm over a 20% sucrose cushion. Following centrifugation, the media and sucrose were discarded and pellets were left to dry for 5 min at room temperature. Pellets were then resuspended over night at 4°C in 500 µL of 1x PBS. The next day, concentrated virions were aliquoted at stored at −80°C.

The titer of our viral stock was determined by plaque assay. Vero E6 cells were seeded into a 12-well plate at a density of $2.5 \times 10^5$ cells per well, and infected the next day with serial 10-fold dilutions of the virus stock for 1 hr at 37°C. Following virus adsorption, 1 mL of overlay media, consisting of 2x DMEM supplemented with 4% FBS and mixed at a 1:1 ratio with 1.2% Avicel (DuPont; RC-581), was added in each well. Three days later, the overlay medium was removed, the cell monolayer was washed with 1x PBS and fixed for 30 min at room temperature with 4% paraformaldehyde. Fixed cells were then washed with 1x PBS and stained for 1 hr at room temperature with 0.1% crystal violet prepared in 10% ethanol/water. After rinsing with tap water, the number of plaques were counted and the virus titer was calculated. The titer of our P2 virus stock was $4 \times 10^8$ PFU/mL.

## SARS-CoV-2 cholesterol depletion assay

The day prior to infection, A549 expressing hACE2 and hTMPRSS2 cells were seeded at a density of $2 \times 10^4$ per well in a poly-L-lysine coated flat-bottom 96-well plate. The next day, MBCD initial stock was prepared at a concentration of 20 mM in 1x PBS and 2-fold serial dilutions were then made using 1x PBS. Prior to infecting cells, a 1 hr pretreatment of MBCD with SARS-CoV-2 virus or cells was carried out. Viral pretreatment was performed by mixing 25 µL of each MBCD dilution with 25 µL of SARS-CoV-2 (MOI of 0.5; $1 \times 10^4$ PFU) per well then incubated for 1 hr at 37°C. For cell pretreatment, media was removed from wells and cells were washed once with 1x PBS. Cells were then incubated for 1 hr at 37°C with 25 µL of each MBCD dilution further diluted in 25 µL of 1x PBS.

Following pretreatments, media or PBS/MBCD mixes were removed from wells and wells were washed twice with 1x PBS. Untreated and MBCD-treated cells were then infected for 1 hr at 37°C with 50 µL of SARS-CoV-2 pretreated with MBCD, or with untreated SARS-CoV-2 (MOI of 0.5), respectively. One hour following virus adsorption, media was removed, cells were washed twice with PBS, and 200 µL of DMEM containing 10% FBS and 1% penicillin-streptomycin was added in each well. Forty-eight hours post infection, cell culture media was harvested and stored at −80°C. Cells were washed twice with 1x PBS, and fixed with 200 µL of 10% neutral buffered formalin for 1 hr at room temperature. Cells were then washed twice with 1x PBS, and taken out of the BSL-3 laboratory.

## SARS-CoV-2 RT-qPCR

To determine SARS-CoV-2 RNA copies, total viral RNA was isolated from cell culture media using a Zymo Research Corporation Quick-RNA Viral Kit (Zymo Research) according to manufacturer's instructions. Viral RNA was quantified using single-step RT-quantitative real-time PCR (Quanta qScript One-Step RT-qPCR Kit; VWR) with primers and Taqman probes targeting the SARS-CoV-2 E gene as previously described (*Corman et al., 2020*). Briefly, a 20 µL reaction mixture containing 10 µL of Quanta qScript XLT One-Step RT-qPCR ToughMix, 0.5 mM Primer E_Sarbeco_F1 (ACAGG TACGTTAATAGTTAATAGCGT), 0.5 mM Primer E_Sarbeco_R2 (ATATTGCAGCAGTACGCA CACA), 0.25 mM Probe E_Sarbeco_P1 (FAM-ACACTAGCCATCCTTACTGCGCTTCG-BHQ1), and 2 µL of total RNA was subjected to RT-qPCR using Applied Biosystems QuantStudio 3 (ThermoFisher). The following cycling conditions were used: reverse transcription for 10 min at 55°C and denaturation at 94°C for 3 min followed by 45-cycles of denaturation at 94°C for 15 s and annealing/extension at 58°C for 30 s. Ct values were determined using QuantStudio Design and Analysis software V1.5.1 (ThermoFisher). For absolute quantification of viral RNA, a 389 bp fragment from the SARS-CoV-2 E gene was cloned onto pIDTBlue plasmid under an SP6 promoter using NEB PCR cloning kit (New England Biosciences). The cloned fragment was then in vitro transcribed (mMessage mMachine SP6 transcription kit; ThermoFisher) to generate a RT-qPCR standard. See Quantification and statistical analysis for details on statistical comparisons.

## SARS-CoV-2 immunofluorescence

Virus-infected cells were fixed in 4% paraformaldehyde for 30 min. The fixative was removed and the cell monolayer was washed twice with 1x PBS. The cells were permeabilized in 1x PBS + 0.1% Triton-X (PBT) for 15 min at room temperature and washed twice with 1x PBS. The cells were blocked in PBT +10% goat serum (v/v) and 1% BSA (w/v) for 1 hr at room temperature before incubating overnight at 4°C with rabbit anti-SARS-CoV nucleocapsid antibody (1:2000 dilution). The cells were then washed five times with 1x PBS and stained with Alexa568-conjugated goat anti-rabbit antibody (1:1000 dilution) in the dark at room temperature for 1 hr. The cells were washed five times with 1x PBS and counterstained with DAPI (1:1000). Images were acquired using the MuviCyte Live Cell Imaging System (PerkinElmer). Six images were captured per well with a 4x objective lens in an unbiased manner.

## Human pathology

Human pathology studies were performed with the approval of the Institutional Review Board at Brigham and Women's Hospital. Clinical autopsies with full anatomic dissection were performed on SARS-CoV-2 decedents by a board-certified anatomic pathologist (RFP) with appropriate infectious precautions. Lung samples were fixed in 10% neutral buffered formalin, embedded in paraffin, sectioned, and stained with hematoxylin and eosin using standard methods. Immunohistochemistry was performed on 4-µm-thick tissue sections following pressure cooker antigen retrieval (Target Retrieval Solution; pH 6.1; Agilent Dako) using a mouse monoclonal antibody directed against TTF-1 (clone 8G7G3/1; Agilent Dako) at 1:200 dilution. Control lung slides were obtained from the BWH Department of Pathology Autopsy Division archives. Glass slides were reviewed by a RFP using an Olympus BX41 microscope, and microscopic photographs were obtained with an Olympus DP27 camera and Olympus CellSens Entry software.

## Quantification and statistical analysis

### Automated fixed cell image analysis

Automated image analyses were performed using MATLAB R2017b (MathWorks). The fraction of cells fused was measured by adaptive segmentation of a maximal intensity projection of the three image channels to delineate nuclei. For each nucleus, area and mean intensity of both nuclear markers in the GFP and mCherry channels was measured. Nuclei with area less than 10 $\mu m^2$ after gaussian filtering and erosion were removed. Remaining nuclei with mean GFP or mCherry channel signal more than 50-digital levels above background were considered nuclear marker-positive. GFP/mCherry double-positive nuclei were designated as part of a syncytium. The fraction of cells fused was calculated per well as the ratio of syncytium- to nuclear marker-positive nuclei. Total nuclei per well was z-score-normalized to the plate negative control wells and considered viable when >-3.

### Heterokaryon co-culture assay compound screens and dose-response analysis

Cell-cell fusion assay was optimized to maximize z-factor, which describes the statistical separation between positive and negative controls and reflects quality of a screening platform (*Zhang et al., 1999*) (see *Figure 3A*, bottom right, for equation and schematic). All statistically significant results are assessed according to the related z-score. The same reproducible assay workflow and statistical cut-offs was used for screening all compound libraries, target compounds, and dose-responses. This statistical approach allows determination of significance relative to a plate's control wells by a cut-off of >3 standard deviations from the mean of the negative control (absolute value z-score >3). For a given experiment, the negative control wells and the positive control wells characterize the statistics of the entire plate (all the test wells). Effective compounds first pass the z-score cut-off for hit determination, and dose-response provide additional confirmation to the original statistically significant hit.

### Cell culture statistical analyses

All data plots and statistical tests were executed using GraphPad Prism (version 8.0.2) for MacOS. For each plot, number of tested biological replicates is indicated in its figure legend. For heterokaryon co-culture assays, statistical significance was assessed for spike variants relative to SARS-CoV-2 wild-type (WT) using ANOVA with multiple comparisons and Bonferroni correction. For pseudovirus blocking assays and SARS-CoV-2 RT-qPCR experiments, significance relative to sham-treated negative control was determined using unpaired two-tailed student's T-tests.

### Fluorescence recovery after photobleaching (FRAP)

Analysis was performed in MATLAB R2017b (MathWorks) by selecting a circular ROI, ~4 $\mu m$ in diameter, co-centered with the photobleached spot as well as a reference unbleached region of similar size for photobleaching correction. FRAP traces were corrected for photobleaching and fitted to a standard exponential decay model of the form: $I(t) = A(1 - e^{-\tau t})$, with the mobile fraction $A$ and the decay time constant $\tau$ being free parameters. Recovery half-life $\tau_{1/2}$ was derived using the relation $\tau_{1/2} = ln(2)/\tau$.

### Cheminformatics analysis

SMILES strings of all analyzed compound libraries were batch-processed in ChemAxon (version 20.8.2) by first correcting each compound to its major tautomeric and protonation state at a physiologically relevant pH of 7.4. A total of 20 physicochemical parameters were calculated for each compound. A Mann Whitney U test was conducted in R software (version 3.4.3, 2017) to assess statistically significant differences between libraries.

The compounds tested in the fusion screen were divided into two libraries: 'non-hits', containing all non-toxic molecules that passed quality control and with a z-score (fusion) >−3.0 (n = 5551); and 'hits', containing non-toxic molecules that passed quality and had a z-score (fusion) <3.0 (n = 163). Both libraries were filtered for empty wells for which no SMILES codes were available (n = 210 in non-hits and one in hits), yielding a final number of 5504 compounds of non-hits and 162 compounds in hits.

A list of GPCR inhibitors, which represent approximately 35% of FDA-approved drugs, was provided by previous work (*Sriram and Insel, 2018*). DrugBank (version 5.1.7) was downloaded and queried with the list to obtain SMILES codes, yielding the final GPCR library (n = 459) for batch processing and analysis in ChemAxon as described above.Box and whisker plots were plotted and linear regression analysis was conducted in GraphPad Prism (version 8.0.2) for MacOS.

## Scaffold and substructure enrichment analyses

Initial scaffold enrichment analysis was conducted in NCGC Scaffold Hopper software (version 1.0). The batch-processed SMILES strings from ChemAxon were input for the hit library (n = 163) to identify most common scaffolds present in this library. To assess enrichment relative to the starting library, the combined hit and non-hit libraries, that is all non-toxic molecules screened in the fusion assay that passed quality control and did not contain empty wells (n = 5714), were input in the software as well. 34 and 981 scaffolds were identified in the hit and starting libraries, respectively.

To identify the compounds containing the enriched scaffolds, SMILES strings from the top-10 enriched scaffolds in the hit group were analyzed in RDKit (version 2020.02.5) substructure search module using Python programming language (version 3.6.12) and IPython (version 7.12.0). Enrichment was assessed by conducting a pooled population comparison of the frequency of a scaffold in the hit library and the frequency of that same scaffold in the starting library. These values were then used to calculate z-scores and two-tailed p-values. Compounds containing statistically significant scaffolds (p-value<0.05) were visually inspected to assess if unique and more complex substructures exist. If identified, those SMILES codes were subjected to another round of substructure search followed by two-tailed p-value calculation as described.

## Bioinformatics

We first acquired the complete set of viral proteins from viruses that infect humans yielding 1,391,780 proteins (data retrieved October 2020) (*UniProt Consortium, 2015*). Next, we filtered for proteins in which two or more transmembrane prediction tools predicted an overlapping transmembrane helix or a transmembrane helix has been experimentally verified, yielding 168,094 proteins (*Käll et al., 2004*; *Sonnhammer et al., 1998*). Of these proteins, we applied a sliding window approach to assess local density of cysteine residues around the transmembrane helices. Specifically, we scanned the thirty-residue regions that lie on the N- or C- terminal sides of each transmembrane helix, using a window size of 20. For each protein, the transmembrane-adjacent window with the highest fraction of cysteine was taken as the protein's cysteine fractional 'score'. The complete set of protein scores is provided in *Supplementary file 2*. To summarize high-confidence hits, we first removed redundancy by filtering for duplicate sequence entries that originated from strain-specific sequence deposition. This final set is provided as *Supplementary file 2*, with high-density hits called out in *Figure 5G*.

In parallel, we acquired the complete set of human proteins (n = 20370) from Uniprot (data retrieved October 2020) (*UniProt Consortium, 2015*). We then similarly filtered for predicted transmembrane proteins, yielding 5182 candidates (*Käll et al., 2004*; *Sonnhammer et al., 1998*). Of these proteins, we applied the same sliding window approach as for viral proteins as described above. The complete set of protein scores is provided in *Supplementary file 3*. We further subjected these putatively cysteine-rich transmembrane proteins to manual filtering to identify 'spike-like' human proteins, which feature cysteine motifs in cytosol and aromatics at the ectodomain-plasma membrane interface. Results are summarized in *Figure 5H* with gene ontology (PantherDB) presented in *Figure 5—figure supplement 1D*.

## Acknowledgements

We thank all Brangwynne Lab members for helpful discussion and critiques and Evangelos Gatzogiannis for help with live cell microscopy. AD wishes to thank the Hargrove lab at Duke University, and particularly Sarah Wicks, for assistance and use of the ChemAxon analysis software, as well as Dr. Brittany Morgan for helpful discussions. This work was supported by Princeton COVID-19 research funds through the Office of the Dean for Research (CPB and AP labs); the Howard Hughes Medical Institute (CPB lab); a Boston University start-up fund and Peter Paul Career Development

Professorship (FD); NIH (GM095467 and HL122531 to BDL; GM134949, GM124072, and GM120351 to IL); Volkswagen Foundation (IL); Human Frontiers Science Program (IL); a Burroughs Wellcome Fund Award for Investigators in Pathogenesis (AP); Longer Life Foundation—RGA/Washington University Collaboration (ASH); postdoctoral fellowship awards from the Uehara Memorial Foundation and JSPS Research Fellowships for Young Scientists (TT); from the SENSHIN Medical Research Foundation (S.S); and from the Natural Sciences and Engineering Research Council of Canada (CCJ).

## Additional information

### Competing interests

Alex S Holehouse: ASH is a consultant for Dewpoint Therapeutics. Clifford P Brangwynne: CPB is a scientific founder and consultant for Nereid Therapeutics. The other authors declare that no competing interests exist.

### Funding

| Funder | Grant reference number | Author |
| --- | --- | --- |
| National Institute of General Medical Sciences | GM095467 | Bruce D Levy |
| National Heart, Lung, and Blood Institute | HL122531 | Bruce D Levy |
| National Institute of General Medical Sciences | GM134949 | Ilya Leventall |
| National Institute of General Medical Sciences | GM124072 | Ilya Leventall |
| Howard Hughes Medical Institute | Investigator lab | Clifford P Brangwynne |
| National Institute of General Medical Sciences | GM120351 | Ilya Leventall |

The funders had no role in study design, data collection and interpretation, or the decision to submit the work for publication.

### Author contributions

David W Sanders, Conceptualization, Investigation, Methodology, Writing - original draft, Writing - review and editing; Chanelle C Jumper, Paul J Ackerman, Dan Bracha, Conceptualization, Investigation, Methodology, Writing - review and editing; Anita Donlic, Devin Kenney, Ivan Castello-Serrano, Saori Suzuki, Tomokazu Tamura, Robert F Padera, Investigation, Methodology, Writing - review and editing; Hahn Kim, Conceptualization, Investigation, Writing - review and editing; Alexander H Tavares, Mohsan Saeed, Resources, Writing - review and editing; Alex S Holehouse, Funding acquisition, Investigation, Methodology, Writing - review and editing; Alexander Ploss, Ilya Leventall, Florian Douam, Bruce D Levy, Supervision, Funding acquisition, Writing - review and editing; Clifford P Brangwynne, Conceptualization, Supervision, Funding acquisition, Writing - original draft, Writing - review and editing

### Author ORCIDs

David W Sanders [ID] https://orcid.org/0000-0002-1835-6895
Saori Suzuki [ID] http://orcid.org/0000-0001-5233-6604
Tomokazu Tamura [ID] http://orcid.org/0000-0003-1395-6610
Alex S Holehouse [ID] https://orcid.org/0000-0002-4155-5729
Alexander Ploss [ID] http://orcid.org/0000-0001-9322-7252
Clifford P Brangwynne [ID] https://orcid.org/0000-0002-1350-9960

### Ethics

Human subjects: Human pathology studies were performed with the approval of the Institutional Review Board at Brigham and Women's Hospital. Clinical autopsies with full anatomic dissection were performed on SARS-CoV-2 decedents by a board-certified anatomic pathologist (RFP) with appropriate infectious precautions.

### Decision letter and Author response

Decision letter https://doi.org/10.7554/eLife.65962.sa1
Author response https://doi.org/10.7554/eLife.65962.sa2

## Additional files

### Supplementary files

• Supplementary file 1. Related to *Figure 4*. Raw data from unbiased drug repurposing screen.

• Supplementary file 2. Related to *Figure 5*. Viral transmembrane proteins with proximal cysteine-rich regions.

• Supplementary file 3. Related to *Figure 5*. Human transmembrane proteins with spike-like membrane proximal regions.

• Supplementary file 4. Related to *Figures 3–7*. Amino acid composition of transmembrane proteins used in this study and cellular targets associated with extensively tested compounds.

• Transparent reporting form

### Data availability

All data generated or analyzed during this study are included in the manuscript and supporting files with the exception of raw imaging data (>400,000 Nikon ND2 files), which is not feasible to post online given its massive size (>1.5 TB). This data is available from the lead contact upon request, assuming the interested party provides a server with sufficient storage capacity. Raw data (computed fusion scores) from the drug repurposing screen is available in Supplementary File 1; bioinformatics, Supplementary File 3.

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
