## [Decision Letter]

**Acceptance summary:**

Your study describes an important new screening method to identify compounds that inhibit SARS-CoV-2 infection and syncytia formation caused by the virus. This method and the finding that cholesterol plays a critical role in these processes are significant advances, particularly given the ongoing pandemic.

**Decision letter after peer review:**

Thank you for submitting your article "SARS-CoV-2 Requires Cholesterol for Viral Entry and Pathological Syncytia Formation" for consideration by *eLife*. Your article has been reviewed by 3 peer reviewers, one of whom is a member of our Board of Reviewing Editors, and the evaluation has been overseen by Vivek Malhotra as the Senior Editor. The reviewers have opted to remain anonymous.

Summary:

This study investigates how SARS-CoV2, the virus that causes COVID-19 enters cells and may damage the lungs of COVID-19 patients; it identifies drugs that reduce cell-cell fusion, which appears to be correlated with disease severity. The authors identified several compounds that target cholesterol homeostasis and reduce viral-host cell membrane fusion and inhibit SARS-CoV-2 infection. Overall, the study is well done, and the screening method may be a useful tool for others, but additional evidence is required to support the claim that the identified drugs work by lowering cholesterol levels.

Essential revisions:

1. Reviewer 2 raises a number of important concerns about the viral infections results shown in Figure 7 (major points 3-5), specifically about the length of infection and the possible toxic effects of MBCD on cells and the virus. The reviewer makes a number of suggestions about how these concerns could be addressed.

2. Reviewer 1 (point 1) says that since some of the drugs identified in the screen are proposed to decrease fusion by decreasing cholesterol levels in the plasma membrane, this should be directly measured. A second-best option would be to measure total free cholesterol levels in the cells, since most free cholesterol is probably in the plasma membrane.

3. Reviewer 3 points out that much of what this study shows about syncytia formation and the role of cholesterol in infection was already known or at least suggested by earlier work (as detailed in the Reviewer's comments in "Recommendations for the authors."). Please address these concerns by citing and discussing the references the Review mentions and putting your findings in context.

4. Reviewer 2 asks for a better description of the fusion assay and more discussion of cells lines were chosen (major points 1 and 2). Please address these concerns.

5. Please provide direct evidence that spike protein is palmitoylated or soften the claim that it is (Reviewer 1, point 2).

*Reviewer #1:*

This study seeks to identify factors that contribute to the formation of fusion-competent SARS-CoV-2 and the formation of pathological syncytia. Both processes remain poorly understood. Syncytia were detected in the lungs of COVID-19 who died as a direct consequence of SARS-CoV-2 infection, suggesting syncytia formation contributes to pathology. To study syncytia formation, a novel cell in vitro co-culture assay was used. It was found that expressing SARS-CoV-2 spike or ACE2 form large synapse-like structure before fusing to form syncytia. The cell fusion assay was used for a high throughput screen to identify drugs that prevent syncytia formation in vitro. Characterization of the effects of the drugs suggests cholesterol is required for viral entry and fusion.

Strengths of the study are the demonstration that syncytia formation may contributes to COVID-19 pathology and the development of a robust, rigorous assay to study syncytia formation in vitro. The screen for drugs that inhibit syncytia formation is well done and the compounds identified are likely to be useful for study the mechanism of syncytia formation. The remainder of the study is weaker, and the significance of the finding less clear. The investigations of the role of spike protein in fusion and how cholesterol affects fusion require some additional work to be convincing. Also, the idea that achievable reductions in cellular free cholesterol levels could significantly reduce spike-mediated membrane fusion, while intriguing, is not strongly supported by the findings.

1. Much of evidence that cholesterol plays an important role in spike-mediated membrane fusion is indirect or negative, like the finding that spike is not in rafts. Perhaps the strongest evidence for a specific role for cholesterol is provided by the experiments with MBCD (particularly the evidence shown in Figures 6I and S7D,E). On the other hand, since cholesterol is roughly 50% of the lipid in plasma membranes, it is not terribly surprising that removal of a large fraction of cholesterol from cells disrupt fusion. While the study suggests lipid-targeting drugs affect fusion by reducing cholesterol levels in the plasma membrane, this has not been directly tested. A demonstration that plasma membrane levels (or even total cellular free cholesterol levels) decrease when the cells used for the fusion assay are treated with drugs that inhibit fusion would significantly strengthen the conclusions of the study and its implications for treatment of COVID-19 with lipid-targeting drugs. Similarly, while it is interesting that treating virus with MBCD reduces viral fusion, it is not clear that modest reductions in viral cholesterol levels, which is probably the best one could hope for in patients, would have any effect on viral fusion efficiency.

2. While the study makes a good case that the transmembrane domain and intracellular portion of spike are necessary for efficient fusion, the claim that the membrane proximal is palmitoylated or that palmitoylation plays a role in spike-mediated fusion is not well supported. There is no direct demonstration that spike is palmitoylated. In addition, the effect of the palmitoylation inhibitor 2-BP is quite modest in U2OS cells and only better in Vero at high concentrations, given that the IC50 for 2-BP is typically 10-15 μM (e.g, PMID: 23631516). Either better evidence that palmitoylation regulates spike should be provided or the statement about the role of palmitoylation in spike-mediated fusion should be softened or removed.

*Reviewer #2:*

1. Not enough details were provided for the U2OS cells and the spike-ACE2 fusion assay in Figure 1. For instance, what are the protein levels of cellular proteases, including furin (Hoffmann, Mol Cell, 2020) and TMPRSS2/4/11D/13 (Zang, Sci Immunol, 2020; Hoffmann, bioRixv, 2020) in these cells? Is this a clonal or pooled ACE2 expression? How much ACE2 is present at the plasma membrane (by flow cytometry)? Where is the fluorescent tag (GFP or iRFP) constructed? All panels in Figure 1 should be quantified to show the percentage of fused cells and average number of nuclei in syncytia. Same with Figure S1 and other fusion assays.

2. Almost all the inhibitor experiments thus far were carried out in U2OS-ACE2 and Vero based cell assays (Figure 3-4). However, for the pseudotyped virus and authentic SARS-CoV-2 infection, the authors switched to A549-ACE2/TMPRSS2 cells, which predominantly utilizes the plasma membrane mediated entry (Hoffmann, Cell, 2020; Shang, PNAS, 2020). For instance, apilimod, which inhibits SARS-CoV-2 at nanomolar range (Kang, PNAS, 2020) is ineffective here (Figure 7B). This is likely not an inherent problem of transformed cell lines but likely alters the entry pathway utilized. The authors should clarify this better in the text and provide references.

3. Figure 7E, the length of infection (48 hr) is too long to properly access viral entry. With multiple rounds of infection, it is possible that newly synthesized viruses produced from MBCD treated cells are less infectious, thereby resulting in reduced viral antigen signals. Fixing or harvesting infected cells at 6-8 hr will yield signals strong enough to be quantified, to better focus on the entry/early stage replication.

4. Improved images are needed for Figure 7F. It seems that MBCD at higher concentrations dramatically reduced the blue DAPI staining, and yet the authors previously showed that it is not toxic to cells (Figure 6H).

5. Figure 7G, how does MBCD treatment affect viral particle integrity? In other words, is the viral RNA still associated/encapsidated within the viral capsid proteins post MBCD treatment? Plus/minus the RNase treatment in the media or ELISA of viral proteins would be good controls to include.

Overall, I think this is a nicely written and timely study. My major problem lies with viral infection experiments performed in Figure 7, which I think should be fixed to strengthen the conclusion of the manuscript.

*Reviewer #3:*

Syncytia formation has been found in the lungs of patients with SARS-CoV2 and appears to be correlated with disease severity. This phenomena is likely due to cell-cell interactions through Spike and ACE2. Using an assay based on one cell expressing S and the other ACE2 this investigation exhaustively screens existing antiviral compounds as well as compounds known to dusrupt cellular endocytic and secretory pathways to identify factors regulating syncytia formation. The ultimate finding of this study is the important role played by plasma membrane cholesterol for syncytia formation and the potential of cholesterol extracting molecules for blocking the syncytia formation.

While this is a very thorough and comprehensive study, almost all its findings have been previously reported either with CoV2 or COV1 or MHV.

1. The critical importance of syncytia formation in lungs for disease severity has been reported for CoV2 by multiple groups ( Tian et al., Hoffmann et al., Graccal et al., Buchreiser et al., ). Some of these references are missing and credit should be given.

2. Liu et al. in 2003 had reported in Nature that SARS-CoV1 S and ACE2 were sufficient to form syncytia and they used a very similar assay as to one reported here. This reference is missing from the manuscript and should be included.

3. Musarrat et al., in 2020 showed that the antiretorviral protease inhibitor nelfinavir blocks CoV2 viral fusion and prevents syncytia formation. This reference is missing from the manuscript and should be included.

4. The important role of cholesterol in b-coronavirus infections was also previously highlighted in 1988 and 1991 by the group of Anderson

and Methylbcyclodextrin was shown by them to prevent syncytia formation. These references are missing and should be credited.

Given all the above, what remains novel in this manuscript is the exhaustive screening of drugs to identify not only those that block the syncytia from forming but also those that don't have any effect. Therefore it may be a useful resource and suitable for publication.

---

## [Author Response]

Summary:This study investigates how SARS-CoV2, the virus that causes COVID-19 enters cells and may damage the lungs of COVID-19 patients; it identifies drugs that reduce cell-cell fusion, which appears to be correlated with disease severity. The authors identified several compounds that target cholesterol homeostasis and reduce viral-host cell membrane fusion and inhibit SARS-CoV-2 infection. Overall, the study is well done, and the screening method may be a useful tool for others, but additional evidence is required to support the claim that the identified drugs work by lowering cholesterol levels.

We thank the editors and reviewers for taking the time to evaluate our manuscript, “SARS-CoV-2 Requires Cholesterol for Viral Entry and Pathological Syncytia Formation”, for publication at *eLife.* We were pleased to hear that reviewers were enthusiastic overall, stating that the study was “well done” and our novel, high-throughput “screening method may be a useful tool for others”. However, in the evaluation summary, it is noted that “additional evidence is required to support the claim that the identified drugs work by lowering cholesterol”. We are of the strong opinion that the data clearly shows that cholesterol reduction inhibits spike-mediated membrane fusion (see Figures 6,7). Moreover, cheminformatics (Figure 4), pharmacological interrogation of steps of secretory pathway (Figure 3), and identified EC50s for hit small molecules together provides a compelling picture that most (if not all) compounds act by altering biophysical properties of the membrane (see Figure 4).

However, we certainly respect reviewers’ stated limitations of our work and eagerly address these in the revised draft. After consulting with colleagues, we recognize that substantial new experimental data, requiring development/optimization of new techniques, would be needed to determine whether the identified lipophilic compounds (Figure 4) act exclusively at the level of cholesterol. For example, it is possible that certain compounds exert similar biophysical effects on the membrane as cholesterol reduction (e.g. altered fluidity), but exhaustive interrogation of mechanism of action for dozens of compounds is beyond the scope of this study. Given the significance of the findings for the ongoing pandemic and *eLife*’s COVID-19 policies (https://elifesciences.org/articles/57162), we feel it is in the public’s best interest to revise the text to highlight limitations of the current study, which we anticipate can be rigorously addressed in future experimental work.

Please note that reviewer comments are numbered (e.g. [R1-1] is shorthand for Reviewer 1, Comment 1) for easy reference across critiques and the evaluation summary (Essential Revisions, [ER]).

We hope the revised manuscript adequately addresses concerns of limitations, context in light of past work, and methodological details, and is now suitable for publication in *eLife*.

Essential revisions:1. Reviewer 2 raises a number of important concerns about the viral infections results shown in Figure 7 (major points 3-5), specifically about the length of infection and the possible toxic effects of MBCD on cells and the virus. The reviewer makes a number of suggestions about how these concerns could be addressed.

Reviewer 2 raises important points/caveats with respect to our work and offers productive suggestions as to supporting experiments that would strengthen our conclusions. Please see [R2-3] to [R2-5] for detailed address of these concerns.

2. Reviewer 1 (point 1) says that since some of the drugs identified in the screen are proposed to decrease fusion by decreasing cholesterol levels in the plasma membrane, this should be directly measured. A second-best option would be to measure total free cholesterol levels in the cells, since most free cholesterol is probably in the plasma membrane.

To address this for the numerous lipophilic hit compounds identified by our screen (Figure 4), substantial new experimental data, based on optimization of new techniques, would be required (see [ER-0]). For example, one (potentially) facile way to measure free cholesterol would make use of toxinbased probes from the Radhakrishnan group (Das et al., 2014; Endapally et al., 2019a; Kinnebrew et al., 2019). However, such labeled probes are not yet commercially available. Thus, adaptation would require recombinant protein and labeling methodologies (Endapally et al., 2019b), which we are not equipped to deploy in a timely manner. Moreover, while providing nuance to our study, these approaches would likely not change our primary conclusions. Nonetheless, we are enthusiastic about the use of such molecular probes in follow-up studies. We note that such methods are mentioned in the current manuscript (page 7 and 9) and provide additional nuance:

Pg7: “Taken together, the data suggests that spike potentially associates with a specific population of cholesterol, which is biochemically distinct from the sphingomyelin-associated lipid complexes enriched in canonical rafts (Das et al., 2014; Endapally et al., 2019a; Kinnebrew et al., 2019). Future studies will be needed to assess the nature of such cholesterol pools, potentially using toxin-based probes that discriminate between free and inaccessible forms (Das et al., 2014; Endapally et al., 2019b), and how each is affected by the identified lipophilic compounds (Figure 4).”

Pg8: “We surmised that the drug repurposing screen identified compounds that act similarly, thus implicating a counteracting plasma membrane property that increases fusion. […] However, the latter possibility is intriguing, in light of extensive literature on anesthetics and membrane mobility (Cornell et al., 2017; Goldstein, 1984; Gray et al., 2013; Tsuchiya and Mizogami, 2013).”

Pg9: “Precedent for this model is provided by raft-independent yet cholesterol-dependent mechanisms of biomolecular clustering essential for influenza infection (Goronzy et al., 2018; Zawada et al., 2016). […] The interplay between oligomerization, palmitoylation, cholesterol association, and membrane dynamics, and how each of these properties are affected by compounds identified in our screen, will require additional methodologies beyond the scope of this study.“

3. Reviewer 3 points out that much of what this study shows about syncytia formation and the role of cholesterol in infection was already known or at least suggested by earlier work (as detailed in the Reviewer's comments in "Recommendations for the authors."). Please address these concerns by citing and discussing the references the Review mentions and putting your findings in context.

We thank the reviewers for drawing our attention to important works that were not cited in the original version of the manuscript – our examination of previous work in the literature was extensive, but we clearly did not properly cite all studies. Such scholastic oversights have been addressed with textual changes throughout the manuscript. Please see [R2-2], [R3-1], [R3-2], [R3-3], and [R3-4] for specifics.

4. Reviewer 2 asks for a better description of the fusion assay and more discussion of cells lines were chosen (major points 1 and 2). Please address these concerns.

We apologize if the original version of the manuscript provided insufficient details regarding the fusion assay and selection of cell lines. We hope that specifics given to [R2-1] and [R2-2] clarify ambiguities for Reviewer 2 and other readers interested in these details.

5. Please provide direct evidence that spike protein is palmitoylated or soften the claim that it is (Reviewer 1, point 2).

Reviewer 1 is correct in noting that we do not specifically show that spike is palmitoylated, basing our conclusions on a palmitoylation inhibitor (2-BP) (Figure 6) and citation of past work on related coronavirus spike proteins with similar cysteine-rich cytoplasmic tails. Text changes were made to soften claims, provide nuance, and highlight alternative possibilities, which are detailed in response to [R1-2].

Reviewer #1:This study seeks to identify factors that contribute to the formation of fusion-competent SARS-CoV-2 and the formation of pathological syncytia. Both processes remain poorly understood. Syncytia were detected in the lungs of COVID-19 who died as a direct consequence of SARS-CoV-2 infection, suggesting syncytia formation contributes to pathology. To study syncytia formation, a novel cell in vitro co-culture assay was used. It was found that expressing SARS-CoV-2 spike or ACE2 form large synapse-like structure before fusing to form syncytia. The cell fusion assay was used for a high throughput screen to identify drugs that prevent syncytia formation in vitro. Characterization of the effects of the drugs suggests cholesterol is required for viral entry and fusion.Strengths of the study are the demonstration that syncytia formation may contributes to COVID-19 pathology and the development of a robust, rigorous assay to study syncytia formation in vitro. The screen for drugs that inhibit syncytia formation is well done and the compounds identified are likely to be useful for study the mechanism of syncytia formation. The remainder of the study is weaker, and the significance of the finding less clear. The investigations of the role of spike protein in fusion and how cholesterol affects fusion require some additional work to be convincing. Also, the idea that achievable reductions in cellular free cholesterol levels could significantly reduce spike-mediated membrane fusion, while intriguing, is not strongly supported by the findings.

We thank Reviewer 1 for their overall enthusiasm for the rigor of our work and the utility of our cell fusion assay. We sympathize with the argument that additional work is needed to convincingly demonstrate the mechanism of action for many lipophilic compounds identified in our high-throughput small molecule screen (Figure 4) (see [ER-0] and [ER-2] response) and their significance for human patient pathobiology. We address specific critiques from Reviewer 1 below.

1. Much of evidence that cholesterol plays an important role in spike-mediated membrane fusion is indirect or negative, like the finding that spike is not in rafts. Perhaps the strongest evidence for a specific role for cholesterol is provided by the experiments with MBCD (particularly the evidence shown in Figures 6I and S7D,E). On the other hand, since cholesterol is roughly 50% of the lipid in plasma membranes, it is not terribly surprising that removal of a large fraction of cholesterol from cells disrupt fusion. While the study suggests lipid-targeting drugs affect fusion by reducing cholesterol levels in the plasma membrane, this has not been directly tested. A demonstration that plasma membrane levels (or even total cellular free cholesterol levels) decrease when the cells used for the fusion assay are treated with drugs that inhibit fusion would significantly strengthen the conclusions of the study and its implications for treatment of COVID-19 with lipid-targeting drugs. Similarly, while it is interesting that treating virus with MBCD reduces viral fusion, it is not clear that modest reductions in viral cholesterol levels, which is probably the best one could hope for in patients, would have any effect on viral fusion efficiency.

We agree with Reviewer 1 that it may not be surprising that cholesterol is important to productive SARSCoV-2 infection. We now pay particular care to citing past studies on related viruses that reached similar conclusions [ER-3]. Nevertheless, we strongly feel that the high-throughput drug screening methodology (and identified hits) are of broad interest to the scientific community, particularly given the novel assay’s ability to test compounds in the absence of BSL-3 capabilities, which are limited to a select few institutions. As detailed in text revisions described in [ER-2], future studies will assess whether all identified lipophilic compounds act by lowering total cholesterol, available free cholesterol, or via a more general impact on membrane fluidity.

2. While the study makes a good case that the transmembrane domain and intracellular portion of spike are necessary for efficient fusion, the claim that the membrane proximal is palmitoylated or that palmitoylation plays a role in spike-mediated fusion is not well supported. There is no direct demonstration that spike is palmitoylated. In addition, the effect of the palmitoylation inhibitor 2-BP is quite modest in U2OS cells and only better in Vero at high concentrations, given that the IC50 for 2-BP is typically 10-15 μM (e.g, PMID: 23631516). Either better evidence that palmitoylation regulates spike should be provided or the statement about the role of palmitoylation in spike-mediated fusion should be softened or removed.

Reviewer 1 is correct in noting that we do not directly show that spike is palmitoylated (e.g. by biochemistry gold standards), relying exclusively on the moderate inhibitory effect of 2-BP and citation of previous works. However, the modest effect relative to other works might be expected, given how rapidly transcellular ACE2-spike synapses and fusion events occur upon co-culture (Figure 1). Not only have we referenced the above paper detailing the IC50 for 2-BP (Zheng et al., 2013), we mention this caveat and provide nuance in response to this critique (see Page 6-7):

Pg6: “Consistent with studies on similar coronavirus spike proteins (Liao et al., 2006; McBride and Machamer, 2010a; Petit et al., 2007), mutagenesis of all spike cysteines to alanine severely diminishes cell-cell fusion in both U2OS and Vero models (Figures 5I-L; Figure 5—figure supplement 1B,C). […] Given the relatively modest and cell type-dependent effect of 2-BP treatment, future work using biochemical approaches will be required to confirm the role of palmitoylation and the precise mechanism by which spike’s aromatic-rich transmembrane domain associates with cholesterol to drive membrane fusion.”

P7: “In light of the important role for palmitoylation in tricellular tight junction assembly (Oda et al., 2020), these findings suggest that SARS-CoV-2 may operate by a similar mechanism to promote adhesion and transcellular interfaces, an exciting possibility to be explored in future studies.”

Pg7: “Thus, SARS-CoV-2 spike protein facilitates membrane-fusion in a manner that could be dependent on palmitoylation of its uniquely cysteine-rich CTD, but through a mechanism unique from canonical membrane nanodomains, although we cannot rule out a discrepancy in lipid raft properties between GPMVs and living cells (Levental et al., 2020).”

Reviewer #2:1. Not enough details were provided for the U2OS cells and the spike-ACE2 fusion assay in Figure 1. For instance, what are the protein levels of cellular proteases, including furin (Hoffmann, Mol Cell, 2020) and TMPRSS2/4/11D/13 (Zang, Sci Immunol, 2020; Hoffmann, bioRixv, 2020) in these cells? Is this a clonal or pooled ACE2 expression? How much ACE2 is present at the plasma membrane (by flow cytometry)? Where is the fluorescent tag (GFP or iRFP) constructed? All panels in Figure 1 should be quantified to show the percentage of fused cells and average number of nuclei in syncytia. Same with Figure S1 and other fusion assays.

We apologize to Reviewer #2 if insufficient details were provided to fully appreciate the experimental design and execution. We now provide additional details in the main text and methods in response to this critique (see [R2-1F]) and address each of these queries in turn below [R2-1A/B/C/D/E].

[R2-1A] Levels of cellular proteases and other proteins

This is an important consideration. We refer to the global proteomic measurements on U2OS cells performed by Beck and colleagues (Beck et al., 2011). The authors provide copy number estimates for the top 7300 most abundant proteins in U2OS cells. They estimate the following copy numbers for proteins of interest:

ACE2: not detected

FURIN: <500 copies per cell

TMPRSS2: not detected

TMPRSS4: not detected

TMPRSS11D: not detected

TMPRSS13: not detected

Other TMPRSS proteins: not detected

Although not used in this study, a similar immortalized and adherent, human cell line (HeLa) benefits from highly quantitative proteomic work (Hein et al., 2015). Similar to the U2OS study, the authors do not detect the above proteins with the exception of FURIN (present at 6 nM). This is consistent with the idea that such proteins are expressed in a cell type-dependent manner, as has been noted by studies from Pohlmann’s group (Hoffmann et al., 2020) and many others.

We provide further details regarding model cell lines upon introducing the assay on page 3/5 and in the methods on page 48 in response to this critique (see [R2-1F]).

[R2-1B] Clonal or pooled ACE2 expression?

All spike and ACE2 clonal lines are pooled (i.e. polyclonal; cells express fluorescently tagged proteins at different copy numbers). Confocal microscopy confirmed that all cells expressed proteins of interest following lentivirus transduction. Please see page 48 of unformatted manuscript, which now contains additional detail.

We provide further clarity to cell line methods by adding text to methods (see [R2-1F]).

[R2-1C] How much tagged protein is on plasma membrane?

We previously developed a fluorescence correlation spectroscopy (FCS)-based calibration mechanism to estimate the concentration of mCherry- and mGFP-tagged proteins in living U2OS cells (Sanders et al., 2020). Based on these calibrations (see Figure S1 of referenced work), tagged Spike was present in most cells at high nM concentrations, whereas ACE2 was present at 1-5 μM. We do not currently have calibrations for iRFP-tagged proteins, but suspect that concentrations are roughly similar considering extent and timing of fusion, as well as lentivirus titers used to produce polyclonal cell populations. Relative expression of different iRFP-tagged variants was controlled for, however.

We attempt to clarify this issue by revising text on page 48 of the unformatted manuscript (see [R2-1F]).

[R2-1D] Where is the fluorescent tag?

The location of fluorescent tags (as well as flexible linkers used) is detailed for all examined chimeric fluorescent proteins in the Key Resources Table (page 42-46 of unformatted manuscript). In almost all cases (including all ACE2 and Spike variants), the fluorescent protein is affixed to the C-terminus (i.e. cytoplasmic side) of the transmembrane protein

No revisions were made to the text in response to this critique.

[R2-1E] Quantification of percentage cells fused?

Rigorous quantification using heterokaryon cell-cell fusion assay is provided for all spike variants in Figure 5 and Figure 5—figure supplement 1. Images in Figure 1 are representative of qualitative experiments. Similar trends were observed for spike variants regardless of fluorescent tag (mGFP, mCherry, miRFP).

However, we are sympathetic to the argument that the beginning of the paper lacks quantification for spike variants. We now add the results of a new experiment to Figure 2B, which quantifies lack of fusion for controls (see page 17).

[R2-1F] Not enough details were provided for the U2OS cells—modified text (applicable to [R2-1A,B,C]).

Pg3: “We generated pooled populations of human osteosarcoma (U2OS) cells, chosen for their flat morphology and lack of critical fusion machinery (Beck et al., 2011), which stably express fluorescently tagged ACE2 or spike (full-length, “FL” vs. receptor-binding domain, “RBD”; see Figure 1A for domain organization), using the B7 transmembrane (“TM”) domain (Liao et al., 2001; Lin et al., 2013) as a control.”

Pg5: “In almost all cases, inhibition of fusion occurred at lower compound concentrations relative to the U2OS assay, possibly due to differences in ACE2 levels between cell lines (e.g. ~1-5 µM exogenous ACE2 in U2OS cells is likely much higher than endogenous ACE2 in VeroE6 cells; see Methods).”

Pg49: “Generation of stable U2OS and VeroE6 cell lines. U2OS cells were selected for this study due to their flat morphology (ideal for live cell imaging), ease of lentivirus transduction, and absence of most proteins critical for SARS-CoV-2 fusion. […] For these variants, localization to plasma membrane and subsequent binding to ACE2 was confirmed (transcellular synapses) and essentiality for membrane fusion was addressed.”

2. Almost all the inhibitor experiments thus far were carried out in U2OS-ACE2 and Vero based cell assays (Figure 3-4). However, for the pseudotyped virus and authentic SARS-CoV-2 infection, the authors switched to A549-ACE2/TMPRSS2 cells, which predominantly utilizes the plasma membrane mediated entry (Hoffmann, Cell, 2020; Shang, PNAS, 2020). For instance, apilimod, which inhibits SARS-CoV-2 at nanomolar range (Kang, PNAS, 2020) is ineffective here (Figure 7B). This is likely not an inherent problem of transformed cell lines but likely alters the entry pathway utilized. The authors should clarify this better in the text and provide references.

Reviewer 2 is accurate in noting that we use different cell models in high-throughput cell-cell fusion (Figures 1-6; U20S-ACE2 and VeroE6) and pseudotyped virus/SARS-CoV-2 infection studies (Figure 7; A549-ACE2/TMPRSS2). Further, the reviewer is correct that our identified EC50s for apilimod in cell-cell fusion assays (U2OS syncytia, Figure 2E: ~15 uM; VeroE6 syncytia, Figure 3—figure supplement 2B: ~5 uM) are higher than reported by the mentioned studies (cited). However, we disagree with the reviewer’s statement that apilimod is ineffective in the A549-ACE2/TMPRSS2 pseudovirus entry assay (Figure 7B). Indeed, we show that apilimod reduces entry by ~50% at high nanomolar concentration (Figure 7B). The incomplete inhibition may reflect multiple pathways of entry (i.e. direct fusion and endocytosis). Discriminating these possibilities will be important in future studies.

For the current manuscript, the following text revisions were incorporated in light of this critique (see page 5 and 8):

Pg5: “For example, apilimod, a promising COVID-19 drug candidate that inhibits PIKFYVE kinase (Cai et al., 2013; Kang et al., 2020; Riva et al., 2020), was particularly potent (Figure 3E), however less so than in the case of infection studies (Kang et al., 2020; Riva et al., 2020).”

Pg8: “Apilimod, a PIKFYVE inhibitor and promising therapeutic in multiple SARS-CoV-2 models (Kang et al., 2020; Riva et al., 2020) including heterokaryon assays tested herein (Figures 3E; Figure 3—figure supplement 2B), inhibited (but did not completely block) entry at nanomolar concentrations (Figure 7B). […] By contrast, 25-hydroxycholesterol, which lowers plasma membrane cholesterol by redirection to the cell interior (Abrams et al., 2020; Im et al., 2005; Wang et al., 2020; Yuan et al., 2020; Zang et al., 2020a; Zhu et al., 2020; Zu et al., 2020), had no effect (Figure 7C).”

3. Figure 7E, the length of infection (48 hr) is too long to properly access viral entry. With multiple rounds of infection, it is possible that newly synthesized viruses produced from MBCD treated cells are less infectious, thereby resulting in reduced viral antigen signals. Fixing or harvesting infected cells at 6-8 hr will yield signals strong enough to be quantified, to better focus on the entry/early stage replication.

We thank this reviewer for her/his comment. We address these concerns below.

As viral entry is the first step of the viral life cycle, inhibiting this critical step prior to the first round of infection will echo through all subsequent rounds of infection over the next 48-hours. The MOI that we used for this experiment (i.e. 0.5) was specifically determined for that purpose, and to allow us to observe MBCD inhibitory effect at 48-hours post infection from an initial pre-treatment step.

Consistently, the MBCD effect on viral particle production was significantly detectable for several tested MBCD concentrations.

Many studies have demonstrated the relevance of pseudotyped systems to study SARS-CoV-2 entry. The SARS-CoV-2 pseudotyping assay in Figure 7B, combined with the extensive evidence in this study on membrane fusion, provide convincing evidence of the role of membrane cholesterol at the step of SARSCoV-2 entry. The purpose of the experiment employing live SARS-CoV-2 was not to demonstrate that MCBD targets virus entry, but rather to evaluate the impact of MBCD on the infectivity of authentic viral particles in a more biologically relevant context.

While we do acknowledge that multiple rounds of infection have already occurred by 48-hours postinfection, and that this could potentially lead to underestimating the inhibitory effect of MBCD, we believe that it altogether provides an accurate picture of the compound’s ability to impact viral propagation from an initial treatment step. Detection of significant viral inhibition at 6-hours postinfection by IHC or qPCR does not mean that there are no remaining viral particles left unaffected and able to further initiate new infection rounds. Additionally, it is unclear as to whether fixing cells at 6hours post-infection using a MOI 0.5 would yield a baseline level of viral infection high enough to accurately quantify MBCD inhibition. Altogether, we are confident that the experimental settings we employed in this experiment were appropriate to answer the biological question under study.

Finally, we do not have concerns about the possibility that newly synthetized viruses produced from MBCD-treated cells may be less infectious. As shown in Figure 7G (lower panel), MBCD pre-treatment of cells does not impact viral particle production regardless of the concentration tested over a 48-hour infection time course. Consequently, in the context of virus-treated cells, temporary exposure of cells to

MBCD during virus adsorption, prior to extensive washing, is unlikely to impact viral particle production.

No text revisions were made in response to this critique.

4. Improved images are needed for Figure 7F. It seems that MBCD at higher concentrations dramatically reduced the blue DAPI staining, and yet the authors previously showed that it is not toxic to cells (Figure 6H).

The reviewer is correct in noting that at high concentration (e.g. >10 mM), MBCD is toxic to cells, and this can be observed as a reduction in DAPI signal/cell number (e.g. 10 mM panels in Figure 7F). However, there is no discrepancy between the findings in Figure 6H (U2OS-ACE2 cell-cell fusion assay; up to 1 mM MBCD tested) and Figure 7F (SARS-CoV-2 infection; up to 10 mM MBCD). It is possible that confusion resulted from use of axes with different ranges? As Reviewer 2 is right that DAPI staining is reduced in 10 mM conditions, we are of the opinion that higher-resolution images are not required to represent the quantitative data reported in Figure 7G.

No changes were made to the text in response to this critique.

5. Figure 7G, how does MBCD treatment affect viral particle integrity? In other words, is the viral RNA still associated/encapsidated within the viral capsid proteins post MBCD treatment? Plus/minus the RNase treatment in the media or ELISA of viral proteins would be good controls to include.

We thank Reviewer 2 for highlighting this important concern: it is conceivable that at very high MBCD concentrations, viral envelopes might disintegrate, dumping their contents into the cell media. For example, one study found this to occur for HIV and SIV at >80 mM (Graham et al., 2003). In contrast, influenza particles remain intact at much higher concentrations (e.g. >50 mM) (Sun and Whittaker, 2003), which is also true for hepatitis B (Bremer et al., 2009), dengue (Carro and Damonte, 2013), and cytomegalovirus (Gudleski-O'Regan et al., 2012). Regardless of virus examined, permeabilization concerns primarily arose at concentrations of an order of magnitude greater than identified for pseudotyped particle entry (Figures 7B) and SARS-CoV-2 infection (Figures 7F,G). We further note that determined EC50s are consistent with studies on other enveloped viruses (e.g. see above referenced studies). Moreover, at concentrations that might perturb viral envelope integrity, cellular toxicity is observed (Figure 7F, 10 mM panels). The reviewer offers thoughtful experimental suggestions for conclusively ruling out permeability artifacts. We discussed with collaborators whether such experiments could be performed in a reasonable time frame, and unfortunately concluded that optimization of new assays (e.g. ELISAs following sucrose-gradient ultracentrifugation) would be prohibitive.

Thus, we have revised the text to highlight the possibility that permeabilization contributes in part to the inhibition observed. The following text changes were made in response to this critique (page 8):

Pg8: “Pre-treatment of cells with millimolar doses of MBCD, which strongly inhibits both coculture syncytia formation and pseudovirus entry, had no effect on infection as determined by RT-PCR and immunohistochemistry (Figure 7F,G). […] However, the molecular basis of this cholesterol-dependent infectivity, i.e. whether it indeed results specifically from cholesterol-dependent spike fusogenicity, or includes contributions from confounding effects such as large-scale virus permeabilization (Graham et al., 2003), remains to be determined.”

Overall, I think this is a nicely written and timely study. My major problem lies with viral infection experiments performed in Figure 7, which I think should be fixed to strengthen the conclusion of the manuscript.Reviewer #3:Syncytia formation has been found in the lungs of patients with SARS-CoV2 and appears to be correlated with disease severity. This phenomena is likely due to cell-cell interactions through Spike and ACE2. Using an assay based on one cell expressing S and the other ACE2 this investigation exhaustively screens existing antiviral compounds as well as compounds known to dusrupt cellular endocytic and secretory pathways to identify factors regulating syncytia formation. The ultimate finding of this study is the important role played by plasma membrane cholesterol for syncytia formation and the potential of cholesterol extracting molecules for blocking the syncytia formation.While this is a very thorough and comprehensive study, almost all its findings have been previously reported either with CoV2 or COV1 or MHV.

We thank the reviewer for their time and kind comments regarding the rigor of our work. With respect to novelty, it is accurate that many of our findings are consistent with those of previous studies, which we reference as appropriate and build upon with new experiments/approaches. With regard to missing citations, we apologize for the unintentional oversights and appreciate the reviewer’s careful attention to ensuring a complete accounting of previous efforts related to our findings on syncytia. This has certainly improved the revised manuscript.

1. The critical importance of syncytia formation in lungs for disease severity has been reported for CoV2 by multiple groups ( Tian et al., Hoffmann et al., Graccal et al., Buchreiser et al., ). Some of these references are missing and credit should be given.

We agree with the reviewer that SARS-CoV-2-associated syncytia have been reported by other groups, and reference the studies mentioned with the exception of Graccal et al. Perhaps this is a typo, as we are unable to find a Graccal study following a cursory search of the literature? If referring to Giacca et al. (*medrxiv* pre-print), we cite this work.

No revisions were made to the text in response to this critique.

2. Liu et al. in 2003 had reported in Nature that SARS-CoV1 S and ACE2 were sufficient to form syncytia and they used a very similar assay as to one reported here. This reference is missing from the manuscript and should be included.

We thank the reviewer for bringing this oversight to our attention. We admit that insufficient care was devoted to citing SARS-CoV-1 studies. The *Nature* manuscript from Farzan, Choe, and colleagues (Li et al., 2003) is now properly referenced.

Please see page 2 for citation of this work.

Pg2: “Pioneering work on SARS-CoV-1 (Li et al., 2003) as well as recent studies on SARS-CoV-2 identified similar syncytia (Buchrieser et al., 2020; Cattin-Ortolá et al., 2020; Hoffmann et al., 2020a; Ou et al., 2020; Papa et al., 2020; Xia et al., 2020; Zang et al., 2020b)…”

3. Musarrat et al., in 2020 showed that the antiretorviral protease inhibitor nelfinavir blocks CoV2 viral fusion and prevents syncytia formation. This reference is missing from the manuscript and should be included.

We thank the reviewer for bringing our attention to this study. This work from the Kousolos lab is now referenced (Musarrat et al., 2020).

Please see page 5 for newly included citation of this work.

Pg5: “Surprisingly, the antiretroviral protease inhibitor nelfinavir was unique in blocking fusion (Figure 3C), a compound whose therapeutic potential was identified by others (Musarrat et al., 2020). Given that other serine protease inhibitors (AEBSF, leupeptin, camostat) lacked efficacy, inhibition by nelfinavir may be related to its proteolysisindependent targets…”

4. The important role of cholesterol in b-coronavirus infections was also previously highlighted in 1988 and 1991 by the group of Andersonand Methylbcyclodextrin was shown by them to prevent syncytia formation. These references are missing and should be credited.

This pioneering work from Anderson and colleagues on related viruses is now cited (Cervin and Anderson, 1991; Daya et al., 1988).

Please see page 9 for newly included citations and discussion of these manuscripts.

Pg9: “Our approach relies on a combination of high throughput screening, quantitative live cell imaging, and viral infection assays, all of which implicate biophysical aspects of the plasma membrane, particularly cholesterol-rich regions, in facilitating spike-mediated membrane fusion. […] Moreover, our high-throughput cell-cell fusion assay not only negates safety concerns associated with pathogenic viruses, but may allow rapid interrogation of the pathogenic nature of newly emerging spike variants that partially evade current vaccines (Davies et al., 2021; Wibder et al., 2021).”

Given all the above, what remains novel in this manuscript is the exhaustive screening of drugs to identify not only those that block the syncytia from forming but also those that don't have any effect. Therefore it may be a useful resource and suitable for publication.

We appreciate that the reviewer agrees that our findings are useful and suitable for publication. Given the numerous novel aspects highlighted by the reviewer, including the exhaustive high throughput screening and living cell imaging capabilities, we could not agree more.

No revisions were made to the text in response to this critique.